# Spatial transcriptomics deconvolution at single-cell resolution using Redeconve

Zixiang Zhou [1,2,3], Yunshan Zhong [1,3], Zemin Zhang [1,2] & Xianwen Ren [1] ✉

Computational deconvolution with single-cell RNA sequencing data as reference is pivotal to interpreting spatial transcriptomics data, but the current methods are limited to cell-type resolution. Here we present Redeconve, an algorithm to deconvolute spatial transcriptomics data at single-cell resolution, enabling interpretation of spatial transcriptomics data with thousands of nuanced cell states. We benchmark Redeconve with the state-of-the-art algorithms on diverse spatial transcriptomics platforms and datasets and demonstrate the superiority of Redeconve in terms of accuracy, resolution, robustness, and speed. Application to a human pancreatic cancer dataset reveals cancer-clone-specific T cell infiltration, and application to lymph node samples identifies differential cytotoxic T cells between IgA+ and IgG+ spots, providing novel insights into tumor immunology and the regulatory mechanisms underlying antibody class switch.

Spatial transcriptomics (ST) technologies provide new tools to identify the cellular organization and interactions of biological samples, which is pivotal to biomedical studies. Multiple ST technologies have been developed and applied to mouse and human brains, lymph node, heart, etc., providing novel insights into cellular communication networks underlying different conditions. However, sequencing-based ST technologies, e.g., the 10x Genomics Visium platform and Slide-seq[1], are essentially of a spot-by-gene matrix structure, needing additional data to provide the cellular information. While the commercial emergence of imaging-based ST technologies, e.g., seqFISH+[2], MERFISH[3], 10x Genomics Xenium[4], and NanoString CosMx[5], provides subcellular resolution, these technologies are limited by low gene throughput, with hundreds of customized genes detected, making their discovery potential unparallel to whole transcriptome-wide spatial technologies. Therefore, integrative analysis of whole transcriptome-wide ST data together with matched single-cell RNA sequencing (scRNA-seq) data is of high significance for biological discoveries.

Multiple effective and efficient algorithms have been proposed for integrative analysis of whole-transcriptome ST and scRNA-seq data. The current algorithms can be categorized to two groups: (1) mapping-based methods, e.g., NovospaRc[6], Tangram[7], Celltrek[8], and CytoSPACE[9], which map single cells to the positions of ST data according to gene expression similarity or related measures; and (2)

deconvolution-based methods, e.g., CARD[10], RCTD[11], cell2location[12], DestVI[13], SpatialDWLS[14], SPOTlight[15], STRIDE[16], CellDART[17], Celloscope[18], DSTG[19], and Stereoscope[20], which try to reconstruct the ST observations by modeling the experimental process as sampling from different combinations of single cells. Mapping-based methods are superior to the current deconvolution-based methods regarding their single-cell resolution as the resolution of current deconvolution methods is limited to tens of cell types. However, mapping-based methods may introduce artificial biases during the mapping process due to the absence of strong constraint on the reconstruction accuracy of the ST observations. It is urgently needed to develop a deconvolution-based algorithm with single-cell resolution to fully release the biological information hidden in ST data.

In this study, we develop an algorithm, named as Redeconve[21], to estimate the cellular composition of ST spots. Different from previous deconvolution-based algorithms, Redeconve introduces a regularizing term to solve the collinearity problem of high-resolution deconvolution, with the assumption that similar single cell states have similar abundance in ST spots. This algorithmic innovation not only improves the deconvolution resolution from tens of cell types to thousands of single cell states, but also greatly improve the reconstruction accuracy of ST data, enabling illustration of the nuanced biological mechanisms hidden in the ST data. Stringent comparison with the state-of-the-art

[1]Changping Laboratory, Yard 28, Science Park Road, Changping District, Beijing, China. [2]Biomedical Pioneering Innovation Center (BIOPIC), Peking University, 100871 Beijing, China. [3]These authors contributed equally: Zixiang Zhou, Yunshan Zhong. ✉e-mail: renxwise@cpl.ac.cn

algorithms including cell2location, CARD, DestVI, CellTrek, NovoSpaRc, and Tangram demonstrates the superiority of Redeconve in terms of reconstruction accuracy, cell abundance estimation per spot, sparseness of the reconstructed cellular composition, cell state resolution, and computational speed. Application to human pancreatic cancer data reveals novel insights into tumor-infiltrating CD8 + T cells, and application to human lymph node data reveals new clues for the regulatory factors of IgA+ and IgG+ B cells.

## Results

### Redeconve: a quadratic programming model for single-cell deconvolution of ST data

Redeconve uses scRNA-seq or single-nucleus RNA-seq (snRNA-seq) as reference to estimate the abundance of different cell states in each spot of ST data (Fig.1a). Different from previous deconvolution methods, Redeconve does not need to group single cells into clusters and then do deconvolution. Instead, Redeconve treats each cell of the sc/snRNA-seq data as a specific cell state serving as reference to estimate the cellular composition of ST data. The direct usage of sc/snRNA-seq data as reference is conceptually direct and computationally efficient, with the potential to handle the heterogeneity of ST data. However, direct usage of sc/snRNA-seq data as reference will introduce a new challenge, i.e., collinearity. That is, multiple single cells have similar profiles of gene expression, prohibiting the accurate estimation of the abundance of individual cell states. We introduce a biologically reasonable heuristic by assuming that similar cells have similar abundance within ST spots, and thus mathematically introduce a regularization term in the deconvolution model based on nonnegative least regression. Solving this regularized deconvolution model by quadratic programming will produce robust estimation of the cellular composition at single-cell resolution for each spot of ST data.

### High accuracy, resolution, robustness, efficacy, and scalability of Redeconve

We applied Redeconve to multiple ST datasets from various platforms (10x Visium, Slide-seq v2, ST, etc.) and compared the performance with other methods. We first compared the consistency of results among different methods at the cell-type resolution based on a human breast cancer dataset. The results suggested that deconvolution-based methods including Redeconve had higher consistency with each other than mapping-based methods (Fig. 1b), indicating the relative superiority and robustness of deconvolution-based methods. This observation is confirmed on additional ST datasets (Supplementary Fig. 1). Different from previous deconvolution-based methods which only reported cell-type-level results, Redeconve can further dictate fine-grained cell states at single-cell resolution (Fig. 1c and Supplementary Fig. 2). On a ST dataset from a human breast cancer sample, Redeconve resolved 249 different cell states from 9 major cell types (Fig. 1c). On a ST dataset from mouse cerebellum, Redeconve resolved 1000 different cell states from 14 major cell types (Fig. 1c). In contrast, the resolution of previous deconvolution methods is limited by the clustering results of sc/snRNA-seq data.

In addition to the robustness and resolution superiority, Redeconve also improves the reconstruction accuracy of gene expression per spot, and the improvement is independent on similarity measures such as cosine similarity, Pearson's correlation, and Root Mean Square Error (RMSE) between the true ST gene expression profile and the reconstructed gene expression vector (Fig. 1d, and Supplementary Figs. 3–4). Redeconve also reached high accuracy of estimated cell abundance (based on a ground truth by nucleus counting, Fig. 1e and Supplementary Fig. 5), and superior computational speed (Fig. 1f and Supplementary Fig. 6). When suitable reference is provided, e.g., matched scRNA-seq data, Redeconve can reach >0.8 cosine accuracy for most ST spots (Fig. 1d). With no suitable reference available (for

example, only snRNA-seq data are accessible for brain samples), Redeconve still outperforms other methods (Fig. 1d). Pairwise comparison between Redeconve and other methods further shows the superiority of Redeconve on almost all spots regarding the reconstruction accuracy (Supplementary Figs. 7–12). Because Redeconve conducts deconvolution analysis spot by spot, parallel computation is enabled and thus Redeconve demonstrates superior computation speed compared with current deconvolution algorithms (Fig. 1f and Supplementary Fig. 6).

To evaluate the performance of Redeconve in estimating the absolute abundance of cells within ST spots, we applied Redeconve to three datasets: Mouse Brain, PDAC and Human Breast Cancer Xenium, in which the cell counts were obtained by nucleus counting based on image segmentation[12,22,23]. Without any priori information, the results of Redeconve showed high conformity with the "ground-truth" cell counts (Fig. 1e), similar to those methods with cell counts (or cell density) as priori knowledge e.g., cell2location and Tangram (Supplementary Fig. 5). We used Shannon entropy to estimate the potential number of different cell states within each spatial spot (see Methods for details about using perplexity as a metric). Redeconve revealed high spot heterogeneity by showing that some spots had complex cellular composition while others had a relatively simple one. In contrast, the entropy of other methods is uniformly high, showing that each spot had been composed of almost all the cell types in reference, which is unrealistic (Supplementary Fig. 13).

### Single-cell resolution is unique to Redeconve compared with previous deconvolution algorithms

Then we examined whether the current deconvolution-based algorithms could be upgraded to single-cell resolution by switching the required cell types to thousands of single cells as Redeconve does. Among all the methods we evaluated, only cell2location and DestVI completed the task but took a rather long time compared with the cell-type inputs (Supplementary Fig. 14) while other algorithms reported errors. Although single-cell inputs improved the reconstruction accuracy of cell2location on the ST data of a human lymph node sample based on the 10x Genomics Visium platform, cell2location did not reach improvement on the human pancreatic tumor and mouse brain datasets, and DestVI failed on all three evaluations (Supplement Fig. 15). In contrast, Redeconve outperformed cell2location and DestVI on almost all spots of the evaluated datasets (Fig. 2a). When switching the inputs from cell types to single cells, DestVI achieved well sparsity regarding the different cell states within each spot (measured by perplexity according to Shannon entropy), similar to the performance of Redeconve. But cell2location reported extremely high perplexity for most spots, indicating overpredicted presence of almost all cell types and thus high false positive rate (Fig. 2b). Therefore, changing inputs from cell types to single cells cannot upgrade the performance of current algorithms to levels parallel to that of Redeconve, and the superiority of Redeconve analysis is mainly derived from algorithmic innovation.

### Evaluating the impact of cell-type resolution on deconvolution by simulation

To evaluate how the cell-type resolution of reference data impacts the deconvolution analysis, we devised a series of simulation experiments to showcase the performance differences of Redeconve and the state-of-the-art algorithms. We constructed three pseudo-bulk RNA-seq datasets by averaging the gene expression data of individual cells based on scRNA-seq data from the PDAC[24], human lymph node[12,25] and human testis[26] datasets separately (Fig. 2c and Methods). Then we applied Redeconve and cell2location, the only alternative method capable of this task. With direct comparison with ground-truth, the results indicate that Redeconve performs substantially better than cell2location, as evidenced by its significantly higher accuracy

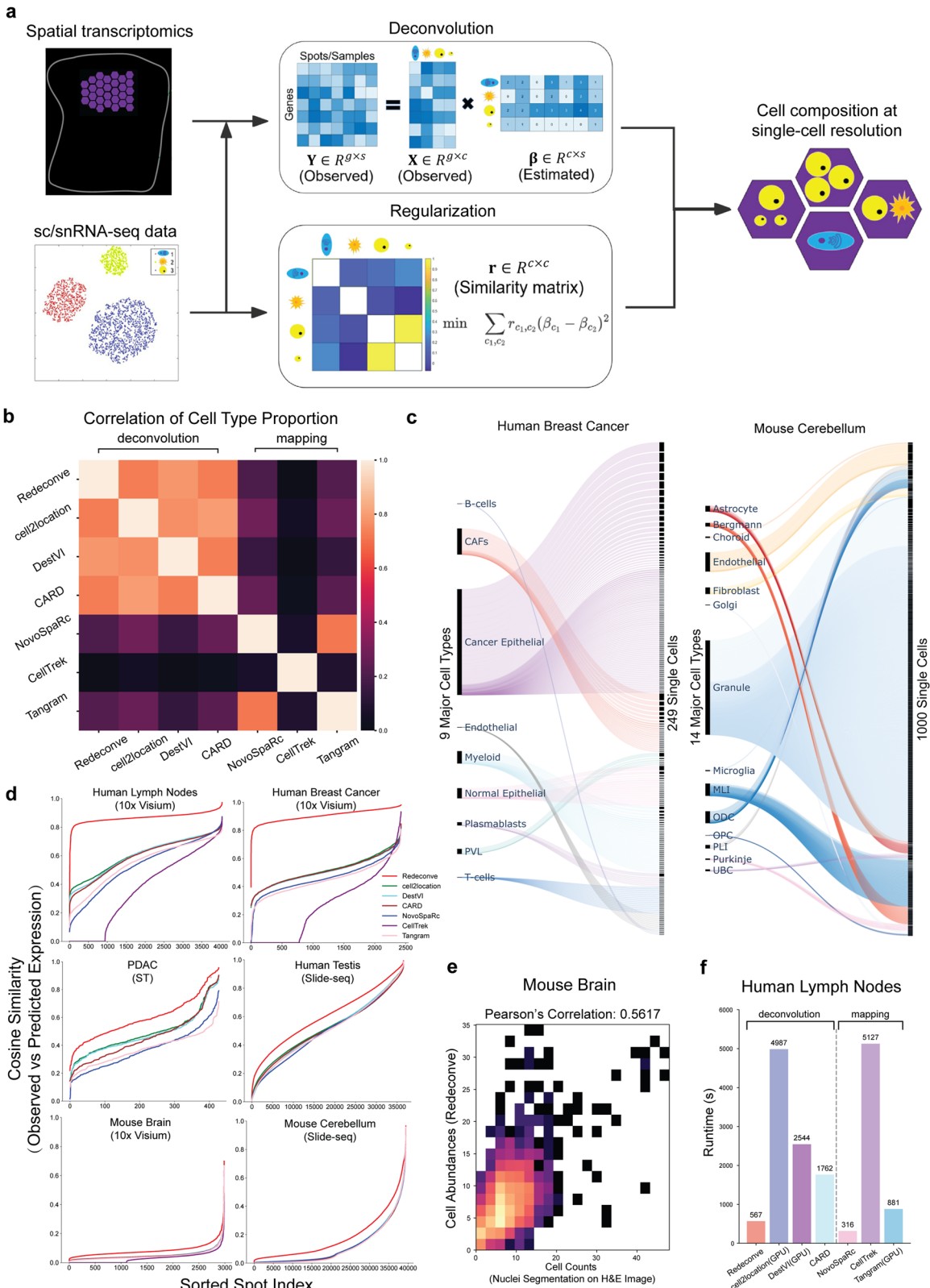

(Supplementary Figs. 16–18). When examining the relationship between accuracy and number of clusters in single-cell reference, Redeconve showed an increase in accuracy when the number of clusters grows, while cell2location experienced a sharp drop (Fig. 2d). This suggests that Redeconve is capable of handling large-scale scRNA-seq data more effectively and can use finer-grained clusters to increase accuracy instead of becoming confused. Furthermore, simulation

experiments also corroborate the validity of using perplexity as a metric of sparsity (Supplementary Table 1 and Methods).

**Evaluating the estimating accuracy of cell-type proportion by 10x Genomics Xenium data as ground truth**

Single-cell ST platforms, such as MERFISH[3], Xenium[4] and CoxMx[5], are commercially emerging as a powerful tool for the high-resolution

**Fig. 1 | Overview of the Redeconve algorithm and benchmark analysis.**
**a** overview of Redeconve workflow for deconvoluting spatial transcriptomics data. Redeconve requires sc/snRNA-seq data together with spatial transcriptomics data as input and performs deconvolution by solving a regularized non-negative least regression model with the aims to estimate cellular composition across spots at single-cell resolution. **b** heatmap illustrated median spot-level Spearman's correlation of cell type proportions among different algorithms on a human breast cancer dataset. **c** Sankey diagram demonstrated the cell-type and single-cell resolutions of Redeconve results on human breast cancer and mouse cerebellum datasets, respectively. The bar height of cell types or single cells refer to their estimated abundance after deconvolution. **d** line chart of cosine similarities between observed and reconstructed expression profiles per spot based on six ST datasets. *N* = 4039, 2426, 428, 36550, 2987 and 39431 spots for human lymph nodes, human breast cancer, PDAC (pancreatic ductal adenocarcinoma), human testis, mouse brain and mouse cerebellum respaectively. Spots were sorted by an ascending order of the cosine similarities. **e** Pearson correlation of cell abundances between Redeconve and the cell counts per spot based on a mouse brain dataset. The ground truth cell counts per spot was obtained by nucleus counting of cell segmentation image[12]. **f** computational efficiency of different deconvolution-based and mapping-based algorithms on a human lymph nodes dataset. Source data of 1c-e are provided as a Source Data file.

mapping of the precise location of single cells, but are limited by the number of genes profiled during experiments because customized probes specific to target genes need to be designed and synthesized before experiments. The high resolution of these platforms provides natural ground truth to evaluate the performance of Redeconve. Here, we used a human breast cancer Xenium dataset generated by 10x Genomics[4] to evaluate the performance of Redeconve regarding reconstruction of ST spot expression profiles, cell type proportion predictions and abundance of individual cell states. This dataset encompasses not only Xenium data containing coordinates and expression profiles of segmented single cells, but also matched scRNA-seq (including 5′, 3′ and scFFPE-seq) and Visium data, enabling us to generate ground truths for Visium spots regarding cell abundances and cell type proportions (See Methods for details). 3906 Visium spots overlapped with the Xenium data were extracted for comparative analysis (Fig. 3a). Compared with the state-of-the-art algorithms including cell2location, DestVI, CARD, NovoSpaRc, CellTrek, and Tangram, Redeconve demonstrated superior cosine similarities between the predicted cell type proportions and the ground truths for most of the Visium spots (Fig. 3b). Specially, Redeconve exhibited superior performance on more than 60% and 70% of spots compared to alternative deconvolution-based or mapping-based methods, respectively (Supplementary Fig. 19). Redeconve, cell2location and Tangram demonstrated comparable performance in estimating the absolute cell abundance within Visium spots, as evidenced by high Pearson's correlation with the ground-truth cell counts indicated by the overlapped cell counts according to the Xenium data, but the performance of Redeconve was more robust to the selection of scRNA-seq references (Fig. 3c and Supplementary Fig. 20). Similarly, the performance of Redeconve in reconstructing the expression profiles of different Visium spots was also more robust to the selection of different scRNA-seq references compared with the state-of-the-art algorithms (Fig. 3d and Supplementary Fig. 21).

## Single-cell resolution by Redeconve enables identification of pancreatic cancer-clone-specific T cell infiltration

To demonstrate the power of deconvolution at single-cell resolution on solving practical biological problems, we further investigated the Redeconve results of the human pancreatic ST dataset[24]. The ST is from the original ST platform, and scRNA-seq data from the same individual were obtained through InDrop. Redeconve with single cells as reference outperformed other methods regarding the reconstruction accuracy for almost all the spots (Fig. 4b, c and Supplementary Fig. 7). Using cell types as reference and varying the cell-type resolution from 20 to 318 clusters, Redeconve still resulted in stable superior performance compared with other methods (with the same inputs) (Supplementary Fig. 22), suggesting the advantage of Redeconve by excluding the interference of single-cell reference vs cell-type reference, although Redeconve is the only algorithm designed to take single cells as reference as we demonstrated in the previous sections. Benchmark regarding individual cell types again showed the superiority of Redeconve. We identified marker genes for each cell type (Supplementary Table 2), and calculated the expression consistency between observed and reconstructed expression profiles per spot based on six ST

between ST observation and reconstructed profiles by different algorithms across all spots (See Methods for details). Redeconve outperformed other algorithms on most cell types (13/20 in top one), especially for cancer, ductal, endocrine cells, and demonstrated comparable performance to the best performers on the remaining of cell types (20/20 in top three, Supplementary Fig. 23a, b). In addition, the performance of Redeconve, cell2location, and Tangram was robust to cell type abundance variations in scRNA-seq data, while the performances of DestVI, CARD, and NovoSpaRc were positively correlated with cell type abundances (*p*-value < 0.05) (Supplementary Fig. 23c).

Histological analysis based on H&E staining identified four tissue regions: pancreatic, cancer, duct epithelium, and stroma[24] (Fig. 4a). Redeconve, CARD, and DestVI successfully distinguished the four types of tissue regions, consistent with histological analysis (Supplementary Fig. 24,). Meanwhile, cell2location, NovoSpaRc and Tangram failed in several conditions (Fig. 4d and Supplementary Fig. 24). Further inspection into a specific spot in the upper cancer region (Fig. 4d, the upper zoomed-in piechart) shows that deconvolution-based methods (Redeconve, cell2location, DestVI and CARD) are able to detect fibroblast, which is known to be abundant in pancreatic cancer[24,27,28], while mapping methods (Tangram and NovoSpaRc) fail in this task.

Then we examined the detailed characteristics of tumor-infiltrating T cells based on these results, which is important to understand the tumor immune microenvironment of pancreatic cancers. The results of cell2location, NovoSpaRc, Tangram and DestVI reported T cells in almost all spots (Fig. 5a), inconsistent with the nature of PDAC as cold tumors; Meanwhile, Redeconve and CARD clearly suggested the sparsity of tumor-infiltrating T cells in pancreatic cancer, consistent with the spatial distribution of T cell-related genes (*CD3*, *IL32* and *TMSB4X*, Fig. 5a, Supplementary Figs. 25–27). As CARD is limited by the cell-type resolution, it is difficult to provide more detailed insights, but Redeconve analysis enables deeper investigation. We identified three T cells in the reference scRNA-seq data that appeared in multiple ST spots, indexed as "T.cell.8", "T.cell.11" and "T.cell.35" separately (Fig. 5b). By examining their expression profiles in the reference scRNA-seq, we identified T cell 11 as regulatory T cell (*CD4*+ *FOXP3*+) and 8 and 35 as *CD8*+ cytotoxic T cells. For fair comparison, we further divided T cells in the scRNA-seq reference data into three groups, i.e., cytotoxic, helper and regulatory T cells and used these three T cell types together with other cell types as reference to re-run other deconvolution algorithms (Supplementary Fig. 28). Consistent with the spatial distribution of *CD8* and *FOXP3*, the result of Redeconve is the most reasonable (Supplementary Figs. 25 and 27). According to the Redeconve deconvolution results, almost all the T cells within cancer region were similar to regulatory T cell 11, and T cell states similar to 8 and 35 only appeared outside or at the edge of the cancer region (Fig. 5b, c), consistent with the immune suppressive status of the cancer region of pancreatic tumors[24,29].

We further conducted co-localization analysis of these three T cell states with the resting cell states by calculating the Pearson correlation coefficient of abundance across all spots based on the Redeconve results (Fig. 5d). The results suggested that the regulatory T cell state similar to T cell 11 mainly co-localized with macrophages similar to

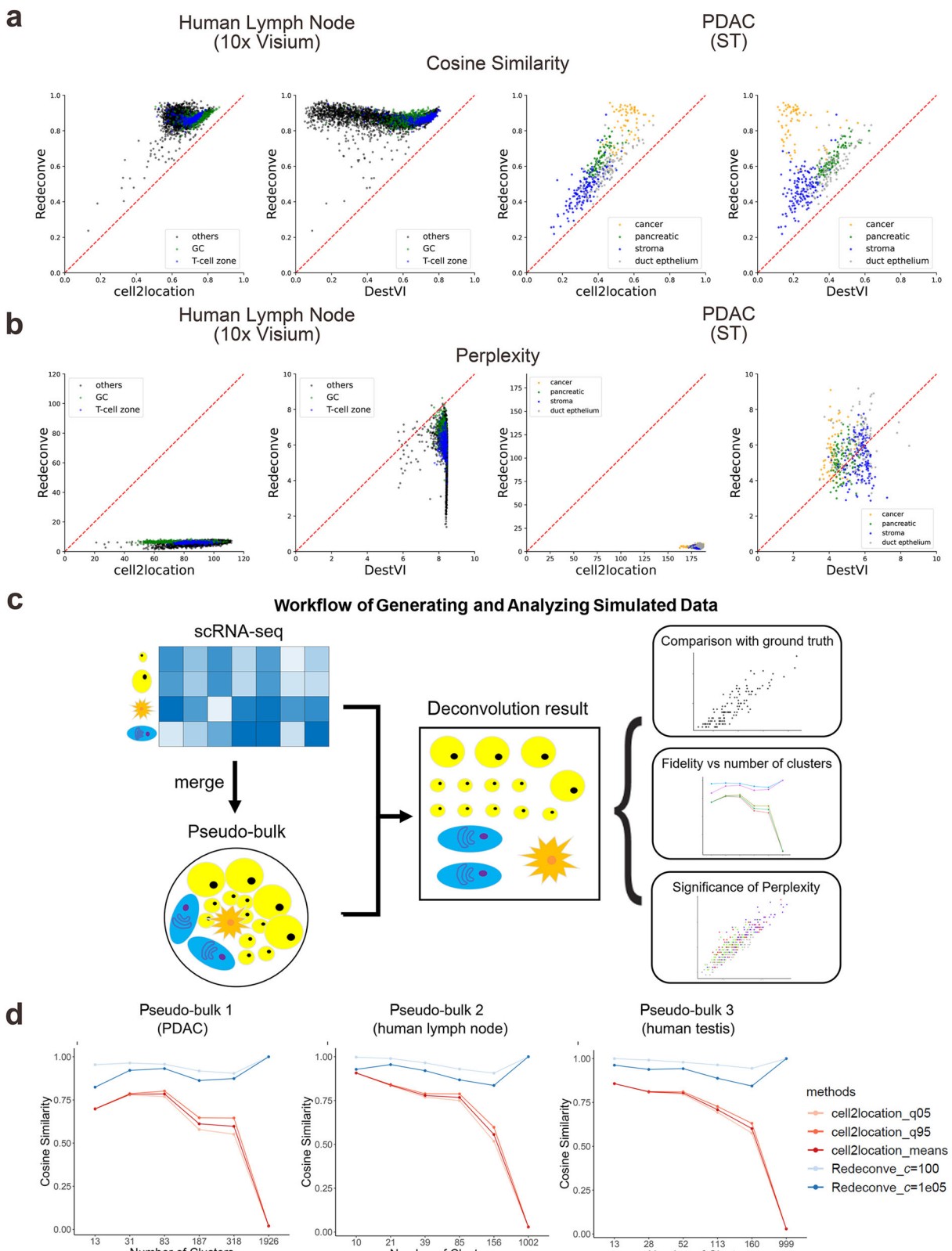

**Fig. 2 | Performance benchmarking with single-cell inputs and simulated datasets.** Redeconve, cell2location and DestVI are currently the only three deconvolution-based tools with the ability to handle thousands of cell states. **a** cosine similarity between true and reconstructed spatial expression profiles based on Redeconve, cell2location and DestVI with 1000 single cells as input. Each dot represents a spot of the ST data. **b** the number of different cell states within each spot estimated by the perplexity of cell state composition per spot for results of Redeconve, cell2location and DestVI with 1000 single cells as input (See Methods for details). **c** workflow of generating simulation data. ScRNA-seq data were aggregated to a pseudo-bulk, which was then used for deconvolution analysis and the results were used for downstream analyses in (**d**). **d** cosine similarity between true and reconstructed spatial expression profiles vs. number of clusters on simulated pseudo-bulk. PDAC, pancreatic ductal adenocarcinoma. Source data of 2a, b and d are provided as a Source Data file.

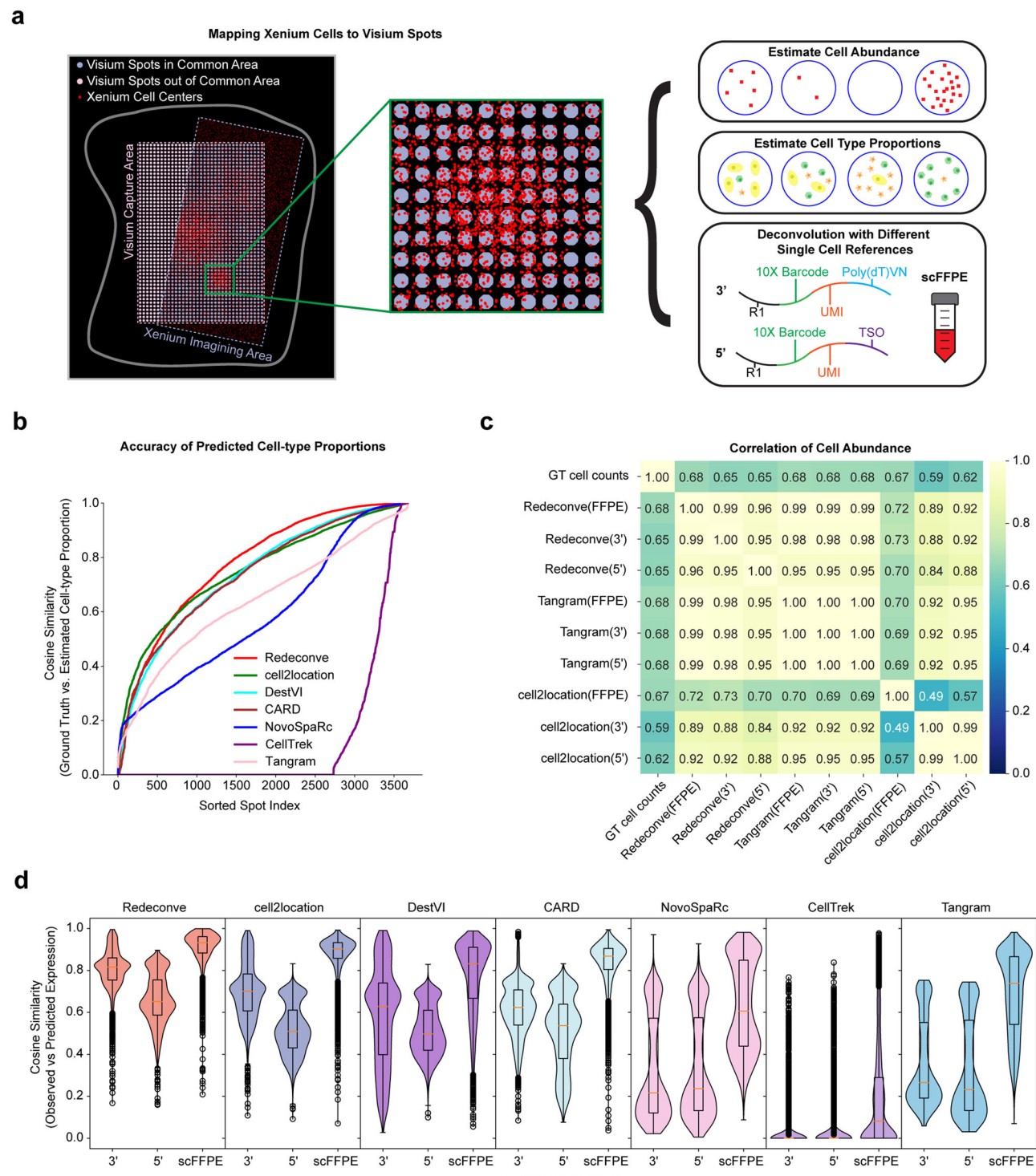

**Fig. 3 | Benchmarking Redeconve performance on a human breast cancer Xenium dataset. a** Left: Overlapped Xenium cells and Visium spots were illustrated on H&E image. Right: the overlapped region was employed for benchmarking Redeconve performance by introducing different single-cell references to predict expression profiles, cell type proportions, and cell abundances. **b** line chart of cosine similarities of cell type proportions between ground truths and algorithm-based predictions per spot. *N* = 3906 spots for the dataset and spots were sorted by an ascending order of the cosine similarities. **c** Heatmap illustrating the pairwise Pearson's correlation of cell abundances among the ground truth, Redeconve, cell2location and Tangram based on various single cell references. **d** violin and box

plot of cosine similarities between observed and reconstructed expression profiles for Redeconve and alternative approaches with different single cell references (3', 5' and scFFPE-seq). The number of independent single cells in the references are 5527, 13,808 and 28,180 respectively. The center line and the bounds of box refer to median, Q1 and Q3 of scores and the whisker equal to 1.5*(Q3−Q1). The minimum and maximum scores refer to Q1-whisker and Q3+whisker. GT, ground truth. scFFPE-seq, single-cell Formalin Fixed Paraffin Embedded sequencing. Source data of 3b-d are provided as a Source Data file. Display items in this figure were manually generated in Inkscape by the authors.

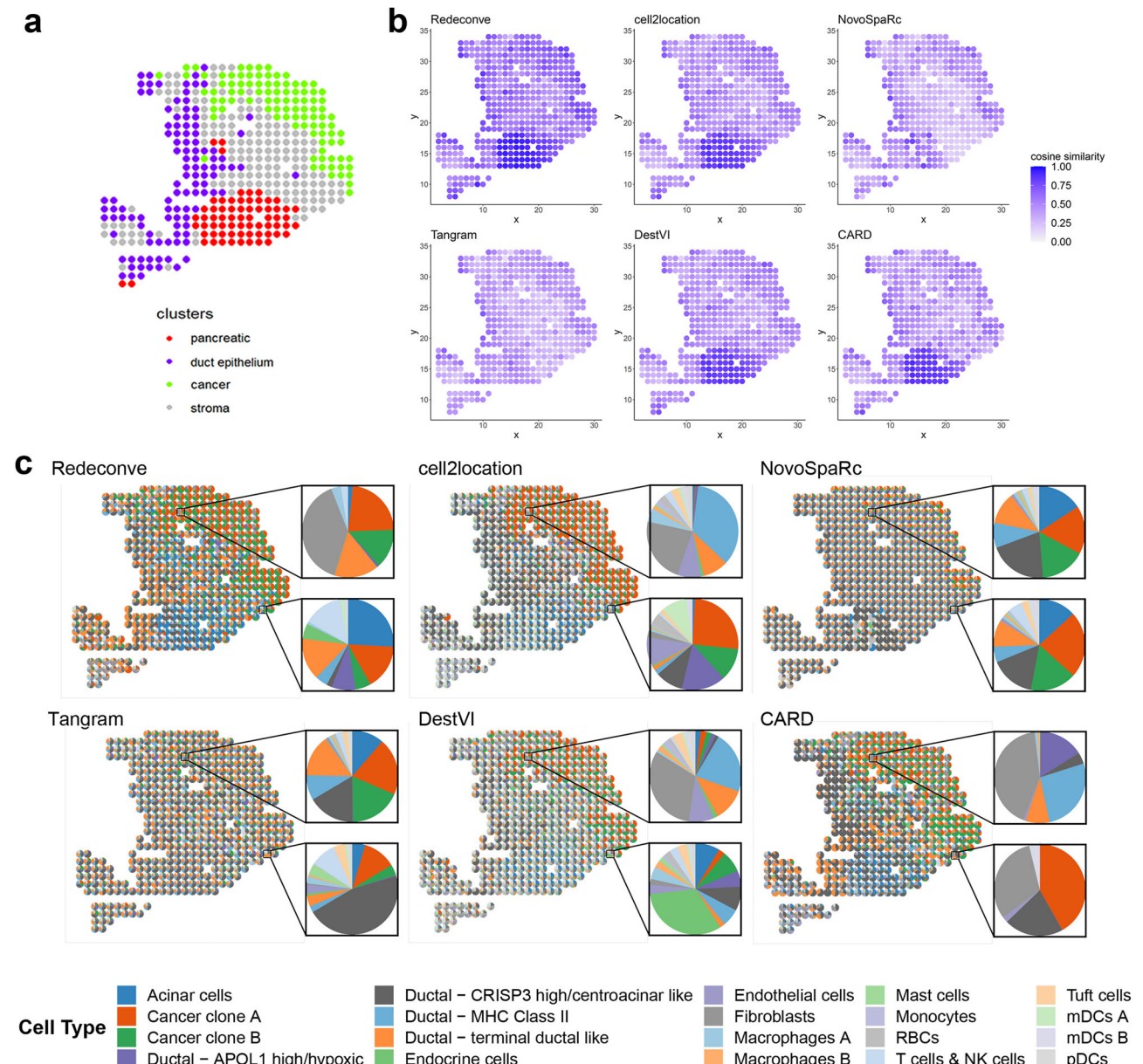

**Fig. 4 | Single-cell deconvolution of a human PDAC (pancreatic ductal adeno-carcinoma) ST dataset. a** four regions were annotated by histological analysis of the original paper: pancreatic, ductal, cancer and stroma regions[24]. **b** spatial distribution of the cosine similarity between true and reconstructed expression profiles per spot by different computational methods. **c** pie charts displaying the spatial distribution of the estimated cell type proportion per spot by different computational methods. RBC red blood cell. mDC myeloid dendritic cell. pDC plasmacytoid dendritic cell. Source data are provided as a Source Data file.

macrophages B. 6, 8, and 16 together with duct cells of two different states. Interestingly, T cell 8 and 35 were mainly co-localized with cancer cells, indicating dispersed cancer cells outside the cancer region. Although provided scRNA-seq reference with higher T cell resolution (cytotoxic/helper/regulatory T cells), such co-localization was not observed by other methods (Supplementary Fig. 29).

Furthermore, these two T cell states were separately co-localized with different cancer clones, with T cell state 8 co-localized with cancer clone B and 35 with cancer clone A. Differential gene expression analysis based on the reference scRNA-seq data further indicated the differences between these two pairs of T cells and cancer cells (Fig. 5e, f). It is revealed previously that *TM4SF1*+ cancer cells denoted late-stage while *S100A4*+ cancer cells (clone B) denoted early-stage[30–32]. Our analysis identified the co-existence of *TM4SF1*+ cancer cells (clone A) and *S100A4*+ cancer cells (clone B) with different *CD8*+ T cells, which is important to understand the interactions between cancer and

T cells. We found that interferon-induced genes (*IFIT1* and *IFI44L*, for example) and HLA-related genes (*HLA-A*, *HLA-B* and *HLA-C*) were all up-regulated in cancer clone B (Fig. 5f), and correspondingly T cell state 8, which is colocalized with cancer clone B, had high expression of *HMGB2*, *HLA-B* and *HLA-C* (Fig. 5f), indicating well-stimulated T cell response[33,34]. In contrast, T cell state 35 was *HMGB2*-negative, *HLA*-low and *TMBS10*-positive and co-localized with more A-type macrophages, indicating a less efficacy state[33,34]. Therefore, with accurate deconvolution at the single-cell resolution, Redeconve can reveal detailed cell-cell interaction at single-cell level and enables discoveries revealing the underlying mechanisms of tumor immunity.

**Redeconve sheds novel insights into the regulatory mechanisms underlying antibody class switch**

Redeconve were further applied to analyze an ST data of human secondary lymphoid organs[12]. We again compared Redeconve with other

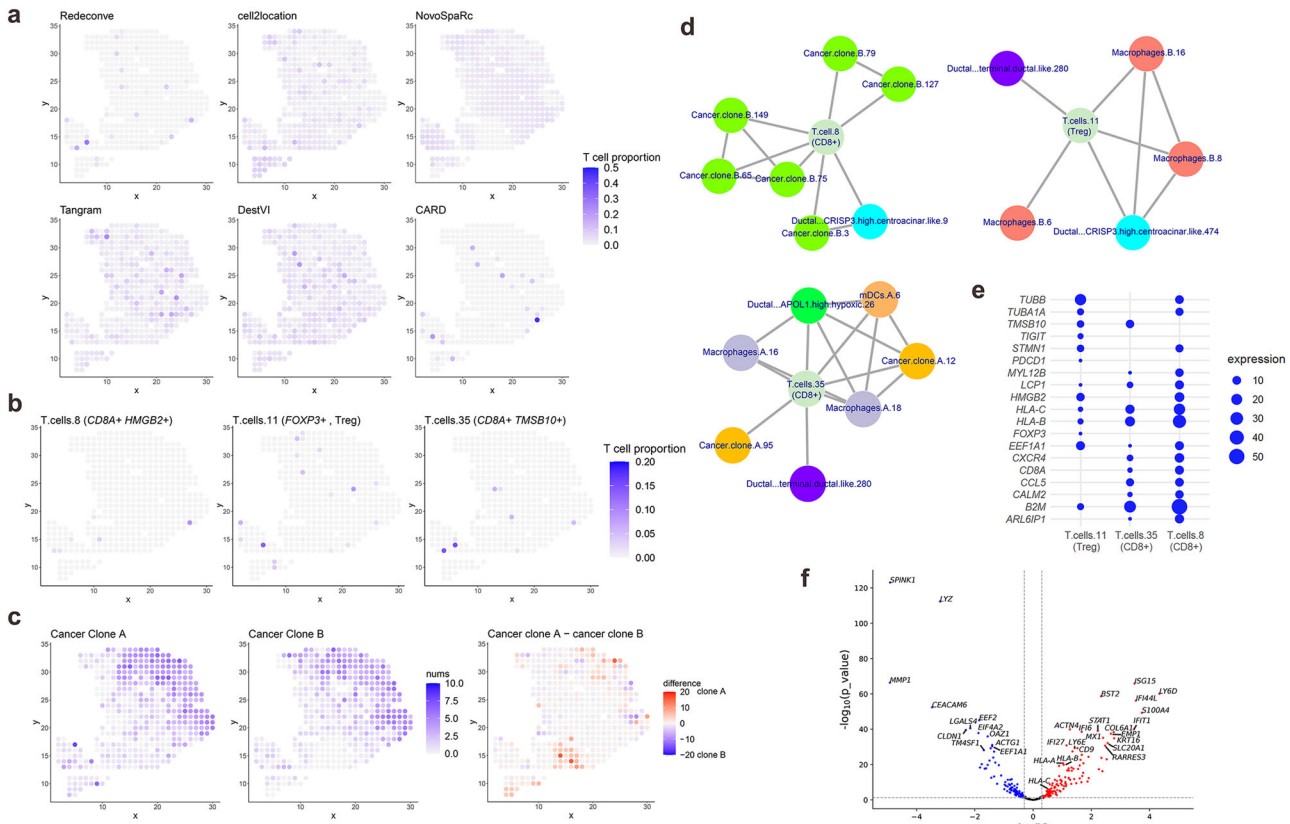

**Fig. 5 | Cancer-clone-specific CD8 + T cell infiltration revealed by Redeconve in human pancreatic cancer. a** abundance of T cells per spot estimated by different methods. **b** single-cell identity of infiltrated T cells revealed by Redeconve. The three T cells are indexed as "T.cells.8", "T.cells.11", "T.cells.35" separately. **c** single-cell identity of different cancer clone cells revealed by Redeconve, together with their abundance difference. **d** co-localization of the three T cell states with other cellular states. Nodes represent single cells and edges represent co-localization (Pearson correlation of cell abundance >0.4). Cancer clone-specific CD8 + T cell infiltration was revealed. **e** dot plot displaying characteristics genes among the three T cell states with different spatial preference with cancer clones A and B.

**f** volcano plot displaying differentially expressed genes between the two cancer clones. The blue and red points refer to up-regulated genes in clones A and B-enriched spots, respectively. Vertical dashed line shows the cutoff of log fold change (±0.3). Horizontal dashed line shows the threshold of -lg p (1.301). T cell response-related genes including interferon-stimulating genes and human leukocyte antigens were up-regulated in clone B-enriched cells. The two-side exact test was applied in edgeR for the statistical test and the *p*-values were calculated without adjustments. Treg, T regulatory. Source data are provided as a Source Data file.

methods on this dataset. In terms of cosine similarity-based reconstruction accuracy, Redeconve achieved mean similarities of 0.868 and significantly outperformed other methods (Fig. 1d). Redeconve achieved high reconstruction accuracy for almost all spots, while, as for other methods, low similarities regions were obvious (Supplementary Fig. 30). We further checked the sparsity of the results by calculating L0-norm. L0-norm of Redeconve has a reasonable distribution between 4 and 32, indicating that only dozens of cell states appear in one spot. In contrast, other methods except CellTrek demonstrated results that almost all cell types appeared in every spot. CellTrek, a mapping-based algorithm, reached low level of *L0*-norm by generating many "zero-cell" spots, of which Redeconve successfully reconstructed the cellular composition (Supplementary Fig. 31).

We further characterized the spatial heterogeneity at single cell resolution to explore the potential regulators of antibody class switch based on this human lymph node data. During the antibody maturation, an activated B cell can change its antibody production from IgM to either IgA, IgG, or IgE depending on the functional requirements, which is termed as class switching[35]. However, the detailed regulators underlying antibody class switching is unclear. Consistent with previous examples, Redeconve outperformed other methods in reconstructing the ST gene expression profiles for almost all spots (Fig. 1d). Spatial pie chart showed that Redeconve produced obvious regional division, while other methods showed blurred or even no boundaries

(Fig. 6a). CellTrek failed to analyze some of the spots. Furthermore, compared with cell-type deconvolution, Redeconve identified 159 different cell states from 17 cell types (Supplementary Fig. 2). 12 different B plasma cell states were identified in the ST data, which can be further divided into 3 groups (IgA + , IgG+ and negative) based on the expression of *IGHA* and *IGHG* genes. Interestingly, we found that IgA+ and IgG+ B plasma cells are spatially mapped to spots in different regions with little overlap, which means that we could define IgA+ and IgG+ spots based on the abundance of those B plasma cells (Fig. 6b). Next, we took one spot in each of the two regions for detailed inspection at the single-cell resolution. The cell proportion of the two spots shows that *CD8*+ T cells account for a large proportion in the IgA+ spot, suggesting latent interactions between *CD8*+ T cells and IgA+ B plasma cells (Fig. 6c). To confirm the universality of such phenomenon, we conducted differential gene expression analysis between IgA + and IgG+ spots to identify up-regulated and down-regulated genes (Fig. 6d). As we expected, *IGHA* and *IGHG* were the most differentially-expressed genes; Genes associated with T cells (*TRAC, TRBC2, CD3D, CD8A* for example) were more up-regulated in IgA+ spots, confirming the existence of such interaction. Since lymph node is one of the organs that generate IgA+ plasma cells, the IgA+ spots might be the potential induction sits for IgA+ plasma cells, and *CD8*+ T cells may play an important role in such process (Fig. 6d). Further co-localization analysis provides more insights (Fig. 6e). We found co-localization of

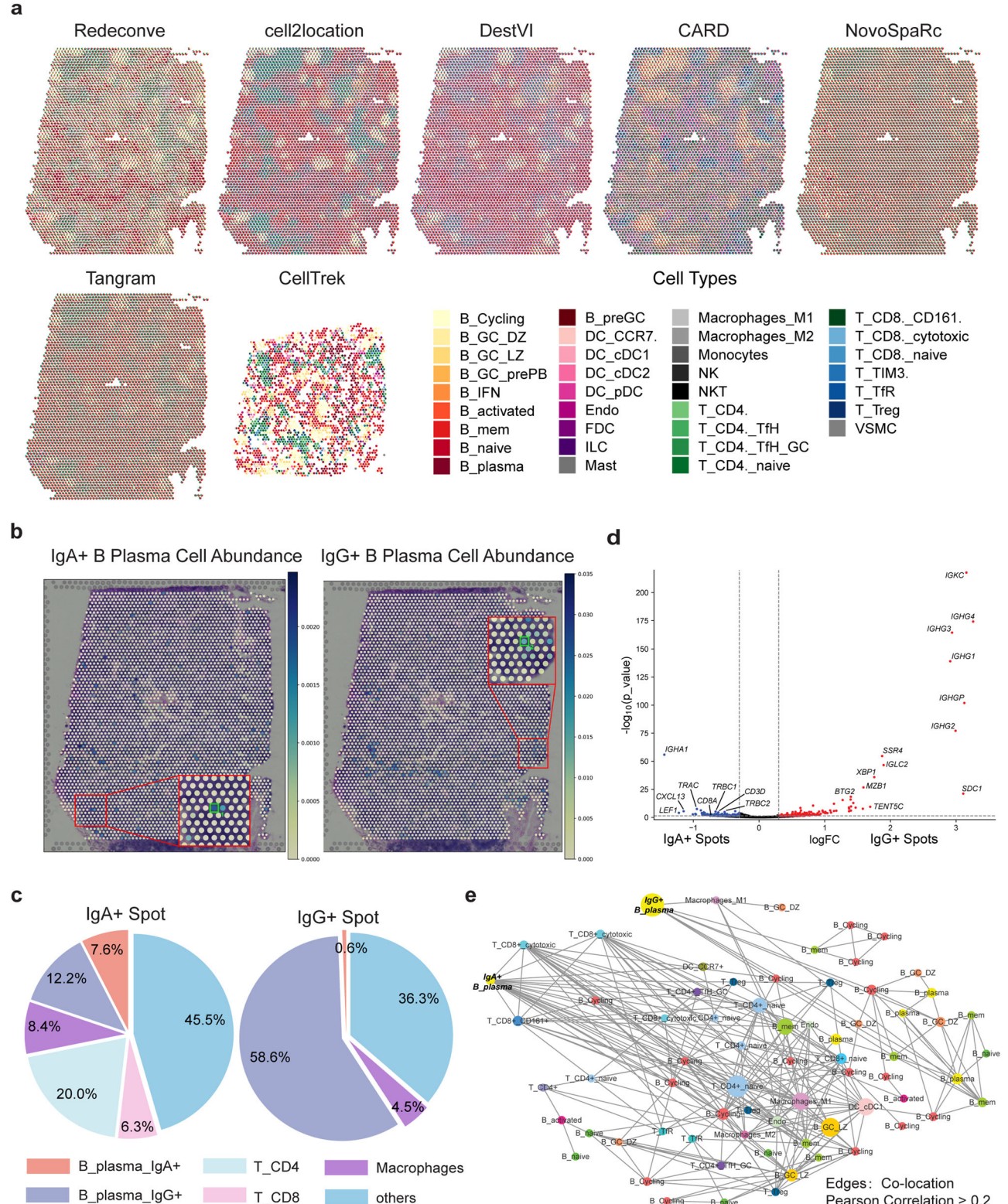

**Fig. 6 | Single-cell deconvolution of a human secondary lymphoid organ ST dataset by Redeconve revealed differences between IgA+ and IgG+ spots regarding cellular composition. a** pie chart displaying the spatial distribution of the estimated cell type proportion by different methods. **b** spatial distribution of IgA+ and IgG+ B plasma cells revealed by Redeconve. **c** comparison of the cell proportion of two selected spots (the IgA+ and IgG+ spots in Fig. 6a with green squares). **d** volcano plots showing the differential gene expression between IgA+ and IgG+ spots. The red and blue point refer to up-regulated genes in IgG+ and IgA+ spots respectively. Vertical dashed line shows the cutoff of log fold change (±0.3). Horizontal dashed line shows the threshold of -lg(p), namely 1.301. The two-side

exact test was applied in edgeR for the statistical test and the *p*-values were calculated without adjustments. **e** co-localization network of IgA+ and IgG+ B plasma cells within the ST data. Nodes represent single cells and edges represent co-located single cells (Pearson correlation of cell abundance >0.2). Abbreviations: GC germinal center, DZ dark zone, LZ light zone, prePB preplasmablast, mem memory, cDC classical dendritic cell, Endo endothelial, FDC follicular dendritic cell, ILC innate lymphoid cell, NK natural killer, NKT natural killer T, TfH T follicular helper, Treg T regulatory, VSMC vascular smooth muscle cell. Source data of 6a-d are provided as a Source Data file.

IgA+ plasma cells with $CD8^+$ cytotoxic T cells, consistent with previous observation that $CD8^+$ cytotoxic T cells can help the formation of IgA+ plasma cells[36,37]. Furthermore, co-location of IgG+ plasma cells and macrophages was identified (Fig. 6e), indicating the roles of macrophages during the genesis of IgG+ plasma cells[38,39]. Hence, deconvolution at single cell resolution by Redeconve gains additional insights that may be helpful for uncovering previously opaque biological question.

## Discussion

Integrative analysis of disassociated single-cell and in situ ST data is pivotal to construct a comprehensive map of the cellular composition and interactomes of tissues. However, because of technological limitations, current computational methods for integrative analysis of single-cell and ST data are limited to the cell type resolution. To deep mine the biomedical information hidden in the single-cell and ST data, here we present Redeconve, a single-cell resolution deconvolution algorithm for integrative analysis of ST data with sc/snRNA-seq data as reference based on a quadratic programming model with regularization of cell-cell similarity, which enables building of comprehensive spatial maps at single-cell resolution for diverse tissues.

We performed stringent evaluation on multiple datasets from a diverse set of ST platforms. The results suggested superiority of Redeconve compared with the state-of-the-art deconvolution-based and mapping-based algorithms in terms of resolution, accuracy, sparsity, robustness, and computational speed. Such improvement from cell-type to single-cell resolution unlocks novel biological discoveries as exemplified by applications in human pancreatic cancer and lymph node samples.

While Redeconve enables deconvolution at single-cell resolution and thus will be a powerful tool for biomedical discoveries, matching between scRNA-seq and ST data appears to be an important factor determining the quality of deconvolution analysis as shown by our evaluation on different tissues (Fig. 1d). Therefore, construction and selection of reference scRNA-seq data according to the specific ST data configuration will be critical in future applications.

Although Redeconve demonstrates superior computational efficacy compared with the state-of-the-art deconvolution algorithms, the single-cell resolution may require extensive computational cost for resolving thousands of cellular states, especially when the cellular throughput of scRNA-seq technologies increases exponentially. Because of the computational complexity of quadratic programming, Redeconve can currently resolve thousands of cellular states based on a standard machine. An enhanced version based on algorithmic innovation or hardware acceleration is needed to handle scRNA-seq datasets of tens of thousands of cellular states.

Deconvolution at single-cell resolution unlocked by Redeconve may also benefit the imputation of ST data with the aid of the rich information in scRNA-seq data. Redeconve has implemented a function to reconstruct the gene expression profiles of individual spots based on the single-cell deconvolution results based on a parsimony principle. The imputed ST data may be more informative to dissect the cellular states of specific tissues.

In summary, we present an algorithm named as Redeconve for conducting deconvolution-based analysis of scRNA-seq and ST data at single-cell resolution. The usage of Redeconve is expected to help mapping the cellular architecture at fine granularity across diverse biomedical situations including tumor, immune, development, neurology, and other health and disease conditions. Applications to human pancreatic cancer and lymph nodes showed the potential of Redeconve to bring completely novel insights due to the single-cell resolution unlocked and the superior technical metrics of Redeconve compared to the current state-of-the-art algorithms. We expect Redeconve will be a useful tool to advance the application of scRNA-seq and ST technologies in diverse research disciplines.

## Methods

### Algorithm

**Model overview.** In general, we apply an improved linear regression model to deconvolute ST data at single-cell resolution. Given a single-cell (or single-nucleus) expression matrix $X$ with dimensions $n_{genes} \times n_{cells}$ and a ST expression matrix $Y$ with dimensions $n_{genes} \times n_{spots}$ as input, Redeconve returns a matrix $\beta$ with dimensions $n_{cells} \times n_{spots}$ indicating the estimated number of each cell in each spot. The goal of our model is to optimize the following loss function for each spot separately:

$$L(\beta) : = \sum_{j=1}^{J} \left( y_j - \sum_{i=1}^{I} x_{ij}\beta_i \right)^2 + c \cdot \sum_{i_1 \neq i_2} R_{i_1,i_2}(\beta_{i_1} - \beta_{i_2})^2 \quad (1)$$
$$s.t. \, \beta_i \geq 0 \text{ for } i = 1, 2, \ldots, I$$

Here $i = 1, 2, \ldots, I$ denotes cells and $j = 1, 2, \ldots, J$ denotes genes. The first term is the traditional Least Square (LS) term and the second term is a regularization term, $c$ is a hyperparameter tuning the weight between the two terms. We will later explain the regularization term in details.

Note that this is a typical quadratic programming problem, so we can rewrite our goal as:

$$\min_{\beta} \frac{1}{2} \beta^T G \beta - d^T \beta \quad (2)$$
$$s.t. \, a^T \beta \geq b$$

Where $G$ is the Hessian matrix, $d^T = (2\sum_j y_j x_{1j}, \ldots, 2\sum_j y_j x_{Ij})$, and $a^T$, $b$ are separately

$$a^T = \begin{pmatrix} 1 & 0 & \cdots & 0 \\ 0 & 1 & \cdots & 0 \\ \vdots & \vdots & \ddots & \vdots \\ 0 & 0 & \cdots & 1 \end{pmatrix}, b = \begin{pmatrix} 0 \\ 0 \\ \vdots \\ 0 \end{pmatrix} \quad (3)$$

So we can efficiently solve this problem with the *solve.QP* function in R package "quadprog".

**The regularization term.** In sc/snRNA-seq data, the collinearity among cells is serious: cells of the same cell type have very similar expression profiles. This problem would lead to instability of coefficients and reduction of efficiency when directly doing linear regression. To solve this collinearity problem, we further include a regularization term into the loss function. By add this term, we aim at stabilizing the coefficients while having minor effect on the residuals.

In the regularization term $c\sum_{i_1 \neq i_2} R_{i_1,i_2}(\beta_{i_1} - \beta_{i_2})^2$, $R_{i_1,i_2}$ is a measure of similarity between cell $i_1$ and $i_2$, which is

$$R_{i_1,i_2} = \begin{cases} r_{i_1,i_2}, & r_{i_1,i_2} > 0 \\ 0, & r_{i_1,i_2} \leq 0 \end{cases} \quad (4)$$

Where $r_{i_1,i_2}$ is the Pearson correlation coefficient between cell $i_1$ and $i_2$. Namely, when the Pearson correlation coefficient is greater than zero, $R_{i_1,i_2}$ is equal to the Pearson correlation coefficient; otherwise $R_{i_1,i_2}$ is zero. So, we manually bring the coefficients of cells whose expression profile is similar closer. By doing this, we can guarantee the robustness and precision of our result.

**Determination of the hyperparameter.** A key point of this model is how to select the hyperparameter: an extremely small hyperparameter will make the regularization term ineffective, while an extremely large one will greatly affect the fitting residuals. An ideal hyperparameter should be as large as possible while affecting the fitting residual as little as possible. Here we offer 3 ways to set the hyperparameter:

1. "default": use the default hyperparameter we set according to the number of cells and genes;

2. "customized": set the hyperparameter arbitrarily by the user;
3. "autoselection": automatically calculate and select the optimal hyperparameter.

In mode "default", we use the following formula to set the hyperparameter:

$$c = c_0 \cdot n_{\text{genes}} / n_{\text{cells}}^2 \qquad (5)$$

Where $c_0$ is a predetermined constant and is set to $10^5$. The idea of this formula is: (1) the LS term is approximately proportional to $n_{\text{genes}}$, so as $n_{\text{genes}}$ increases $c$ should synchronously increase; (2) the regularization term is approximately proportional to the square of $n_{\text{cells}}$, so as $n_{\text{cells}}$ increases $c$ should decrease by $n_{\text{cells}}^2$.

In mode "autoselection", we apply the following method to determine the optimal hyperparameter:

1. We first calculate a hyperparameter $c_d$ according to the formula in mode "default", and set up a series of hyperparameter $c_1, c_2, c_3, c_4, c_5$ as $0.01c_d, 0.1c_d, c_d, 10c_d, 100c_d$;
2. Then we run deconvolution with these hyperparameters separately, and calculate the residual $\varepsilon_i$ for each $c_i$;
3. We further calculate:

$$d_i = \frac{\Delta \varepsilon}{\Delta c} = \frac{\varepsilon_{i+1} - \varepsilon_i}{c_{i+1} - c_i} \qquad (6)$$

4. We check these $d_i$, then choose $c_i$ that maximizes $d_i$ as the optimal hyperparameter (This indicates: if the parameter continues to increase, the residual will increase significantly). Namely, we choose $c_i$ that satisfies:

$$\max_{i \in 1,2,\cdots,I} d_i = \frac{\varepsilon_{i+1} - \varepsilon_i}{c_{i+1} - c_i} \qquad (7)$$

By this procedure, we can get the hyperparameter that maximizes the power of regularization term while having minor effect on the LS term.

We use examples to illustrate the effect of hyperparameters on the results. We applied Redeconve to the human lymph node dataset with a series of different hyperparameters from 0 to 1e08, then calculated the deconvolution residuals (RMSE_normal) to evaluate the effect of hyperparameter (Supplementary Fig. 32). The results showed that an optimal hyperparameter can enhance the deconvolution precision in addition to avoiding co-linearity caused by closely similar cell states. Also, the hyperparameter would also affect the number of cell states selected in the result. A bigger hyperparameter would lead to more cell states selected (Supplementary Fig. 33). We set the hyperparameter as 0 and 1e04 separately on the PDAC dataset. With a hyperparameter of 1e04, more T cells were detected than a hyperparameter of zero in the PDAC dataset (Supplementary Fig. 34). Considering the distribution of CD3+ cells (Shown in Supplementary Figs. 25–27), this example clearly illustrates how the hyperparameter enables biological discovery.

## Data preprocessing

To run the deconvolution, the following data preprocessing steps are necessary. Note that some steps are alternative according to users' needs.

1. Get the expression profiles of cell type/Sampling of single cells. If a cell-type deconvolution is to be run, we will estimate the expression profile $\bar{x}_{ij}$ of cell type $i$ and gene $j$ as the average expression of gene $j$ across all cells within cell type $i$. If a single-cell deconvolution is to be run and the number of single cells is overwhelming, we will take stratified samples of cells by cell type to get a rational number of cells.

2. Gene filtering. Deconvoluting with tens of thousands of genes is time-consuming or even misleading, so we select highly variable genes before deconvolution for computational efficacy. Filtering criteria include the following three standards: (1) These genes appear in both sc/snRNA-seq data and ST; (2) The variance of these genes in sc/snRNA-seq data must be larger than a threshold (default is 0.025); (3) The average counts per spot must be bigger than a threshold (default is 0.003). This finally results in ~8000 genes for deconvolution. Redeconve allows deconvolution without gene filtering with higher computational cost.

3. Normalization of reference. We add a pseudo-count of 0.5 to the "zeros" in sc/snRNA-seq data, and normalize sc/snRNA-seq data to TPM (transcripts per million). Preprocessing operations are not needed for ST data.

## Real datasets for benchmarking

**PDAC.** ST data of a human pancreatic ductal adenocarcinomas (PDAC-A) with 438 spots and sample-matched scRNA-seq data (InDrop) with 1926 single cells across 20 cell types were integrated by Moncada et al., and an intersection of 19,736 genes was used in our study. The annotation of four main structural regions based on histological analysis by Moncada et al. was used during our analysis to depict the spatial characteristics of the ST data.

**Human lymph node.** Human lymph node Visium data were downloaded from the 10x Genomics website (https://www.10xgenomics.com/resources/datasets/human-lymph-node-1-standard-1-1-0), which includes a total number of 4035 spots. ScRNA-seq data were collected from Kleshchevnikov et al, of which 73,260 cells across 34 cell types were collected. Since this scRNA-seq dataset captured a wide spectrum of immune cell states spanning lymph nodes, tonsils and spleen, we used it as reference to reveal the phenotypic diversity of immune cells when deconvoluting at single cell resolution.

**Mouse cerebellum.** The DropViz scRNA-seq dataset were generated by Saunders A. et al. and were collected by Cable D. M. et al. along with the annotations of the cells. The Slide-seq mouse cerebellum data were collected by Cable D. M. et al. using the Slide-seq v2 protocol[11]. Both of these datasets were downloaded from https://singlecell.broadinstitute.org/single_cell/study/SCP948/robust-decomposition-of-cell-type-mixtures-in-spatial-transcriptomics#study-download.

**Human breast cancer.** Human Breast Cancer Visium data related to the Wu et al. study[40] was available at https://zenodo.org/record/4739739#.Ys0v6jdBy3D. Sample 'CID4290' that includes 2426 in tissue spots was used for deconvolution. ScRNA-seq data that includes 100,064 single cells with annotations (Access number: GSE176078, the NCBI GEO database) served as reference to do deconvolution analysis.

**Human testis.** The processed Human Testis Slide-seq dataset was download from https://www.dropbox.com/s/q5djhy006dq1yhw/Human.7z?dl=0 and sample 'Puck5' with 36,591 spots was used for evaluation in this study[41]. The reference scRNA-seq data that includes 6490 single cells was obtained from the NCBI GEO database with access number GSE112013, and the corresponding annotations were available in the supplementary information Table S1 by Guo et al.

**Mouse brain.** 10x Visium and snRNA-seq data (includes annotation) were available in the ArrayExpress database with accession numbers E-MTAB-11114 and E-MTAB-11115, respectively[12]. Sample 'ST8059048' containing 2987 spots was used for evaluation in this study, and all 40,532 single cells across 59 cell types served as reference. In addition, the corresponding data of nuclei counts estimated by histological image segmentation based on deep learning s was downloaded from

https://github.com/vitkl/cell2location_paper/blob/master/notebooks/selected_results/mouse_visium_snrna/segmentation/144600.csv.

**Human breast cancer xenium.** The Human Breast Cancer Xenium dataset is available at https://www.10xgenomics.com/products/xenium-in-situ/preview-dataset-human-breast. A single FFPE tissue block was analyzed by scFFPE-seq, Visium and Xenium. In addition, 3' and 5' gene expression data from dissociated tumor cells is also available[4].

### Comparing Redeconve with alternative methods
We compared Redeconve with recently developed deconvolution-based methods (cell2location, DestVI[13] and CARD[10]) as well as mapping-based methods (NovoSpaRc, CellTrek[8] and Tangram[7]).

**Criteria of selecting alternative methods.** In considering which methods to include for the comparison, we required methods that (1) are specifically designed for end-to-end estimating the abundance/proportion of cells or cell types using scRNA-seq and ST data as input; (2) demonstrate superior performance in the corresponding publications and third-party evaluation papers; and (3) are peer reviewed with a publicly available software implementation before Dec 2022.

**Parameter setting.** Prediction results for the 6 datasets were obtained by running the corresponding programs of the algorithms aforementioned based on the default settings except some special considerations: (1) 1000 cells were randomly selected in NovoSpacRc to avoid large number of total cells; (2) 1000 stratified samples of cells were used for Redeconve in almost all the datasets except PDAC where we used total 1926 cells; (3) minCountGene and minCountSpot of the createCARDObject function were set to 0 to prevent unexpected gene or spot filtering in CARD. The output of each method was either a cell-by-spot matrix represented absolute abundance (Redeconve, Tangram) or proportion (NovoSpaRc) of single cells existing at each spot or estimated cell-type abundance (cell2location) or proportion (DestVI, CARD) matrix except CellTrek, of which the outcome was predicted spatial coordinates for individual cells. Hence, for CellTrek, we obtained cell-by-spot abundance matrix by assigning single cells to specific spots according to whether the spot area designed by ST platforms covered the predicted coordinates. We only evaluated CellTrek on the two 10x Genomics Visium-based datasets (human lymph node and human breast cancer) because of running errors on other ST datasets in our computational environment.

**Calculating performance metrics.** To demonstrate superior performance of Redeconve, we firstly estimated predicted expression profiles for spatial spots. For all datasets, spot-wise cosine similarities, Pearson's correlations and RMSEs between observed and predicted spot-by-gene expression matrix were calculated. In order to compute these metrics based on the output of each algorithm, we calculated the predicted expression matrix through two ways: (1) for Redeconve, NovoSpaRc, CellTrek and Tangram, we multiplied spot-by-cell abundance or proportion matrix by the cell-by-gene sc/snRNA expression matrix; (2) for cell2location, DestVI and CARD, we multiplied the cell-type abundance or proportion matrix by the reference cell-type expression matrix, where the reference was generated through averaging sc/snRNA expression data according to cell types. When calculating RMSEs, the total number of UMIs for each spot in both observed and predicted expression profile was normalized to $n_{\text{genes}}$. We then estimated sparsity of the results through calculating cell-type proportion matrices of all programs and comparing the results according to cell-type information entropy and $L_0$ norm. The $L_0$-norm represents number of cell types present at each spot (nonzero values). We also evaluated the performance of cell abundance estimation by Pearson's correlation between results of individual methods (Redeconve,

cell2location, CellTrek and Tangram) and the cell numbers estimated by histological image segmentation based on deep learning for the mouse brain dataset. Finally, computational efficiencies were estimated through comparing total time spent by each algorithm on a computer with Intel(R) Xeon(R) Platinum 8253 CPU, where we set the maximum number of cores to 96. In addition, we tested the run time of these programs on a single NVIDIA A40 card if GPU acceleration supported (cell2location, DestVI, NovoSpaRc, and Tangram).

**Assessment at single cell resolution.** Cell-by-spot abundance matrix is required for comparison among deconvolution-based methods at single-cell resolution. We, therefore, applied Redeconve with 1000 single cells sampled from the reference scRNA-seq data for the two ST datasets (PDAC and human lymph node) and assigned every single cell a unique cell type since cell2location, DestVI and CARD only support cell-type deconvolution. The result matrices of Redeconve, cell2location and DestVI (no result was available for CARD because of running errors) was obtained according to the corresponding default settings. Cosine similarity, information entropy, perplexity and run-time efficiencies were evaluated as mentioned above.

**Information entropy and perplexity.** We calculate Information entropy $H$ and perplexity $P$ for each spot separately by the following formula:

$$H = -\sum_i \beta_i \log_2(\beta_i) \tag{8}$$

$$P = 2^H \tag{9}$$

where $i = 1, 2, \ldots, I$ denotes different cell states. $\beta_i$ were normalized in advance so that their sum equaled to 1 (i.e., they denote proportion rather than absolute abundance). When $\beta$ is uniformly distributed (namely $\beta_i$ is a constant, $\frac{1}{I}$, for all $i$), we can know by simple calculation that the perplexity equals to the number of states $I$. This means that perplexity can reveal the number of states when the distribution is uniform. For other distributions, perplexity can also approximately represent the number of states. "Number of states" in the setting of single-cell deconvolution refers to "number of cell states (or types)". Namely, the perplexity of each spot can approximately represent the number of cell states/types occurred in this spot. By calculating perplexity on simulated and real datasets, we have verified that perplexity showed good consistency with number of non-zero cell types/states in the result, but poor consistency with absolute cell abundance (Supplementary Fig. 35 and Supplementary Table 1).

**Cell-type level benchmark based on the PDAC dataset.** Marker genes were first identified for each cell types (Supplementary Table 2). Then, for each cell type, similarities of marker genes expression between ST observation and reconstructed profiles by different algorithms across all spots were calculated. Ranks of cosine similarities of individual cell types were used as metrics to summarize the overall performance. In addition, linear regression and statistical test were used to show relationship between cell type abundances and performance metrics.

**Cell abundance of ST spots on PDAC dataset.** To generate ground truth of cell abundance for each ST spot, we first registered H&E and fluorescent images using Adobe Photoshop CC. Such registration enabled the determination of spatial coordinates for ST spots. After that, Cellpose[13] was applied through squidpy[14] to detect cell nuclei from the H&E image. Finally, we counted the absolute number of nuclei within each spot and referred to these values as cell abundance.

## Generating and analyzing simulation datasets

We used 3 scRNA-seq data to generate simulation data separately: PDAC, human lymph node and human testis. Prior to analysis, all scRNA-seq data were down-sampled to around 1000 cells, with the exception of PDAC which contained a total of 1926 cells. To generate a pseudo-bulk for subsequent deconvolution, all single-cells were aggregated together and assigned an abundance value of 1. To perform deconvolution, we clustered the scRNA-seq reference with 5 different resolutions using FindCluster() function in Seurat package. Together with directly using all single-cells as input, this results in 6 groups of references. Then the differently annotated references were used for deconvolution by Redeconve and cell2location and the results were used to compare with ground-truth, calculate cosine similarity and perplexity (Fig. 2c, d and Supplementary Figs. 16–18).

## Benchmarking on human breast cancer Xenium dataset

The Human Breast Cancer Xenium dataset contains scRNA-seq, Visium and Xenium data for a single FFPE tissue block. By mapping Xenium cells to Visium spots, it becomes possible to generate ground truth data regarding cell abundances and cell type proportions. To achieve this, we chose Replicate 1 of Xenium data to align spatial locations of Xenium cell centers to corresponding H&E images through translation and rotation. After that, a key-point registration approach was employed to align H&E images in Xenium and Visium data based on 155 manually identified landmark features on commonly shared microstructures. Then, FindHomography() function in cv2 package with RANSAC method was applied to transform Xenium to Visium coordinates. Hence, the ground truths of cell abundance were generated through counting the transformed cell centers located within each Visium spot. To further generate ground truths of cell type proportion for Visium spots, we labeled each cluster in scFFPE-seq and Xenium data with a corresponding cell type designation (Supplementary Table. 3–4). The proportions of various types of Xenium cells in Visium spots were considered as ground truth cell type proportions.

Based on the generated ground truths, we computed spot-wise cosine similarities between predicted and ground truth cell type proportions for Redeconve and alternative methods. In this approach, we chose scFFPE-seq data as reference for the deconvolution. In addition, Pearson's correlation was applied to measure the performance of cell abundance estimation for Redeconve, cell2location and Tangram. Finally, a selection of distinct single-cell references (including 5', 3', and scFFPE-seq) were applied for the purpose of assessing robustness of the computational algorithms.

## Downstream analyses after Redeconve deconvolution

**Human lymph node.** We firstly ran Redeconve on default setting to obtain deconvolution result at single-cell resolution. Then, we investigated the spatial distribution of plasma cells after grouping these plasma cells into IgA + , IgG+ and others based on the expression of *IGHA1*, *IGHG1*, *IGHG3* and *IGHG4*. IgA+ and IgG+ spots were determined by the following three steps: (1) identifying the top 50% spots with the highest abundance of IgA+ and IgG+ plasma cell enriched, which were named as spot sets A and G; (2) identifying the difference sets between A and G, and naming as AD and GD; (3) selecting spots from AD and GD with the top 1% IgA+ and IgG+ plasma abundance, which were assumed to be IgA+ and IgG+ spots respectively. EdgeR[42] was applied to perform differential gene expression analysis and identified significantly differential genes between IgA+ and IgG+ spots. Then, we calculated Pearson's correlation coefficient among single cell states in the reference across IgA+ and IgG+ spots and took single cells as nodes and correlated cells (Pearson > 0.2) as edges to generate the cell-cell colocation network.

**PDAC.** We ran Redeconve with all the 1926 single cells as reference, and all the parameters were kept default. For downstream analyses, we first compared Redeconve with existing tools as described in the aforementioned sections. Then, to study the distribution of T cells, we distinguished from NK cells T cells by the expression of *CD3D*, *CD3E* or *CD3G* in the scRNA-seq data. We further picked out those T cells that frequently appeared in the ST spots (T cells 8, 11, and 35). To study the spatial colocalization of these T cells with other cells, we calculated the Pearson's correlation of cell abundance across spatial spots, and generated a colocalization network of single cell resolution using those cell pairs whose Pearson correlation were greater than 0.4 with the R package igraph[43].

## Statistics and reproducibility

For all datasets except for PDAC, we down sampled the sc/snRNA-seq reference to around 1000 cells. Stratified sampling was performed when cell types are available, otherwise simple random sampling was performed. The exact number of chosen cells for each dataset are as follows: human breast cancer: 1001, human lymph nodes: 1000, human testis: 999, Mouse Brain: 1003, Mouse cerebellum: 1003, human breast cancer Xenium (scFFPE): 1001, human breast cancer Xenium (3'): 998, human breast cancer Xenium (5'): 1002. The seed was set to 2233. All other parts of this study do not involve randomization. The Investigators were not blinded to allocation during experiments and outcome assessment.

## Reporting summary

Further information on research design is available in the Nature Portfolio Reporting Summary linked to this article.

## Data availability

All relevant data supporting the key findings of this study are available within the article and its Supplementary Information files. The PDAC data used in this study are available in the Gene Expression Omnibus database under accession code GSE111672. The processed human lymph nodes Visium data are available at 10x Genomics website [https://www.10xgenomics.com/resources/datasets/human-lymph-node-1-standard-1-1-0]. The processed human lymph nodes scRNA-seq data are available from Kleshchevnikov et al. [https://cell2location.cog.sanger.ac.uk/browser.html]. The mouse cerebellum data used in this study are available in the Single Cell Portal database under accession code SCP948 [https://singlecell.broadinstitute.org/single_cell/study/SCP948/robust-decomposition-of-cell-type-mixtures-in-spatial-transcriptomics#study-download]. The processed human breast cancer Visium data are available at zenodo [https://zenodo.org/record/4739739#.Ys0v6jdBy3D]. The processed human breast cancer scRNA-seq data used in this study are available in the Gene Expression Omnibus database under accession code GSE176078. The processed human testis Slide-seq data are available at dropbox [https://www.dropbox.com/s/q5djhy006dq1yhw/Human.7z?dl=0]. The processed human testis scRNA-seq data used in this study are available in the Gene Expression Omnibus database under accession code GSE112013. The processed mouse brain Visium data used in this study are available in the ArrayExpress database under accession code E-MTAB-11114. The processed mouse brain snRNA-seq data used in this study are available in the ArrayExpress database under accession code E-MTAB-11115. The processed Visium, 3' scRNA-seq, 5' scRNA-seq and scFFPE-seq for human breast cancer Xenium dataset are available at 10x Genomics website [https://www.10xgenomics.com/products/xenium-in-situ/preview-dataset-human-breast]. Source data are provided with this paper.

## Code availability

The codes used to generate the figures in this paper is available at https://codeocean.com/capsule/1351962/tree/v1. The package is available on GitHub with detailed documentation at https://github.com/ZxZhou4150/Redeconve, https://doi.org/10.5281/zenodo.8384152[21].

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

## Acknowledgements

This work was supported by Changping Laboratory, the National Natural Science Foundation of China (32022016 X.R., 92159305 X.R., and 31991171X.R.), National Key R&D Program of China (2020YFE0202200 X.R. and 2022YFC3400904 X.R.).

## Author contributions

X.R. conceived this study, designed the algorithm, supervised the analysis, and wrote the manuscript. Z.X.Z developed the software, conducted the data analysis, and wrote the manuscript. Y.Z. conducted the data analysis and wrote the manuscript. Z.M.Z provided valuable discussion on the data analysis and wrote the manuscript.

## Competing interests

The authors declare no competing interests.
