## [Peer Review File · Nature Communications]

Spatial transcriptomics deconvolution at single-cell resolution using RedeconveReviewer #1 (Remarks to the Author):

Reconstructing single cell composition of lower than single cell resolution spatial transcriptomics data is an active area of research and currently a largely unresolved problem in computational biology. The work presented by the authors is therefore very timely and any advancements in this area would be very welcomed by the research community. Existing methods for ST deconvolution use cluster-level information to infer the composition of individual ST spots, assuming that a cluster represents a homogeneous, representative cell type. Here, the authors describe a novel method that instead deconvolutes ST spots based on transcriptomes of single cells instead. While conceptually novel and potentially a very useful approach, I have a number of comments/concerns that should be addressed before I can recommend the manuscript for publication.

Implementation:

- The implementation of the described algorithm is not particularly user friendly or well documented. Publishing this as a package with adequate documentation would be essential. I would also highly recommend providing a Seurat/scanpy wrapper as this would make the method much more accessible and more highly used.
- The manuscript and the documentation provided with the code implies that the output of the deconvolution returns a matrix of estimated cell counts per spot. While testing this with a 10x Visium and paired scRNA-seq dataset, the output was neither cell counts (floats, not integers) nor proportions per spot (do not add up to one). Some clarification on this – or better documentation – is required. The results otherwise seemed largely sensible, so I don't think I have grossly mis-used this.
- Unlike other methods where deconvolution outputs a much smaller cell type matrix, the output here generates a much larger cell by spot matrix which is not readily interpretable without additional downstream analysis, and the output cannot be visualised as a whole beyond small toy or heavily down sampled datasets. Again, expanding the current code and providing this as an analysis package with additional functionality for querying, visualising and summarising the output would go a long way to enhance the usability of this. Currently, there is quite a lot of additional legwork required from the user just to understand the basic output.
- Throughout the manuscript, the authors claim that their method is fast ("direct usage of scRNA-Seq data as reference is computationally efficient") in comparison to others– however, in the discussion they concede that it is not very scalable. Indeed, testing deconvolution of one Visium section and a very heavily down-sampled (50 cells per cluster) scRNA-Seq reference dataset – the running time was close to 8 hours. Attempting deconvolution of a single Visium slide using a medium sized (~50k cells) reference scRNA-Seq dataset did not finish (yet). While compared to algorithms like cell2location, which is particularly slow, this is indeed an improvement, many others do take a fraction of the time. As scalability with reference dataset size is an issue, providing some formal guidance on strategies on how to best deal with this for end users with larger datasets would be useful. I noted that there is a section in the methods regarding dataset down sampling – it would be particularly useful to see how much impact, if any, this has on the results.

Benchmarking:

- In many areas, the benchmarking against other methods is very much stacked in favour of Redeconve and the comparisons are not fair.
- While I understand that one of the advantages of this method lies in the ability to map reference dataset cell states to ST spots regardless of the quality and resolution of annotations in the reference dataset, nonetheless I do not think it is fair to feed such high level annotations as "T-cells" to other algorithms and then claim that the method proposed here can resolve the spatial location between Tregs and cytotoxic T-cells and competitors cannot. The authors should repeat this benchmarking and give the other methods a fair chance by using better annotated reference datasets – indeed, as Tregs and CD8s are such functionally different cells, any reasonable researcher would not group them together for a deconvolution task.
- To follow up on a previous point, many existing spatial deconvolution algorithms do not perform well when given a scRNA-Seq reference dataset annotated with closely transcriptionally related reference cell populations, often splitting the prediction probabilities between cell clusters with highly correlated transcriptomes. An algorithm to accurately map finer reference cell sub-populations to spatial transcriptomics data is therefore sorely needed and Redeconve approach does look promising in this regard. However, a benchmark would be needed to evaluate this – it is not explored at all at what reference dataset cluster resolution do other methods break down, but

it would be a very useful reference point to have and highlight niches where the proposed method would be most applicable. Instead the authors do the opposite and restrict other methods to high level annotations only.

- Cosine similarity is used as a performance metric. The difference between the proposed method and others appears to be quite variable dataset-to-dataset, and also between spatial regions. It is likely that this arises due to how the predicted spot transcriptomes are computed – in the case of other deconvolution algorithms, this is based on the mean expression profile of the cell population. Therefore, if the reference dataset is under-clustered, the cosine similarity will likely show the greatest divergence in those cases and is not a reflection of real inaccuracies.
- The authors should also consider benchmarking their method using alternative metrics, for example using simulated single cell mixtures. Cosine similarity alone as a metric is very high level and may hide performance issues for certain cell types – a small fraction of cell type specific genes deviating in the predicted transcriptome would not have a large impact on the overall transcriptome similarity, but would have big implications for data interpretation.
- Alternatively, the authors should consider using some of the single cell resolution spatial transcriptomics data available publicly – e.g. Cosmx or Merscope – as ground truth. For example, mouse brain sections are made available by both 10x and Vizgen, and therefore it would be interesting to use histologically equivalent regions to compare deconvolution results with close to “ground truth” single cell spatial data.
- Why did the authors select some but not other algorithms to benchmark against? An exhaustive comparison is probably beyond the scope of this work, but some justification of the current selection would be good.
- In several instances the authors claim that their algorithm yields better delineation of spatial region boundaries. “Redeconve and CARD showed relatively clear boundaries of tissue regions in accordance with histological analysis (Fig. 3d). Meanwhile, cell2location and DestVI showed blurred boundaries, and NovoSpaRc and Tangram did not show boundaries (Fig. 3d).” This is somewhat subjective – to me, it seems that cell2location does quite well in this instance, though it is interesting to see that NovospaRc and tangram seem to completely fail at this particular task. “Spatial pie chart showed that Redeconve produced obvious regional division, while other methods showed blurred or even no boundaries (Fig. 5c).” In this case, it also looks like cell2location and desIV give quite clear regional divisions. The authors should formalise and quantify these interpretations - for example, by computing cell type composition cluster purity within histologically defined regions.
- The amount of space dedicated to robustness comparisons is unnecessary and somewhat misleading – comparing a deterministic algorithm with non-deterministic ones in such a way is not particularly meaningful, although the poor stability of some other non-deterministic methods is interesting. In figure 2, the values on the scales should also be standardised as at first glance, given the small axis labels, this gives a false impression of much poorer stability of cell2location than it actually is.
- Comparisons attempting to use other methods with single cell input is interesting, but also ultimately not particularly useful, as these algorithms were not designed to work that way.
- “Interestingly, T cell 8 and 35 were mainly co-localized with cancer cells, indicating dispersed cancer cells outside the cancer region, which is missed by other methods”. There is no formal comparison done between Redeconve and other methods on how well their predictions are able to reconstruct co-localisation networks. This statement is based on the lack of inference of T-cell subpopulations by other methods, however as mentioned previously, this is based on the artificially restricted reference dataset annotation level. A comparison of cell co-localisation networks would be particularly interesting, especially as other methods tend to falsely infer co-localisation for closely related cell types.

Other:

- When focusing on specific cell predictions and case study sections, the authors should show the spatial expression of cell type specific genes as well as the signal obtained using their method as a “sanity check”. This is lacking through-out the manuscript. For instance, the authors imply that other algorithms erroneously predict some T-cell presence in the majority of spots in the cancer tissue slide, while Redeconve yields more localised predictions. Showing whether this corresponds to spatial expression of T-cell specific genes such as CD3D, etc. would go a long way towards convincing the reader this is indeed the truth.
- “We found co-localization of IgA+ plasma cells with CD8+ cytotoxic T cells, suggesting that

CD8+ cytotoxic T cells may play important roles during the formation of IgA+ plasma cells. Furthermore, the co-location of IgG+ plasma cells and macrophages may indicate the roles of macrophages during the genesis of IgG+ plasma cells (Fig. 6d).” Co-localisation does not necessarily mean there is a functional role in class switching. I think these statements should be toned down or placed in context of relevant citations supporting these interpretations.

- Similarly, a lot of the more biological examples discussed in the manuscript are poorly cited throughout and lack appropriate literature context.
- The analysis against nuclei segmentation is quite nice – it would be great to repeat it with different resolution/spot size ST platforms to evaluate whether this result is robust across a range of real world values, or merely tuned to perform well at the most common cell densities encountered with current gen ST platforms.
- The manuscript does not convey very well how the hyperparameter tuning affects the predictions. For instance, how does this affect closely related cell type localisation. Some additional exploration and guidance to the end user would be useful.
- Following on from previous point, is Redeconve conceptually really mapping single cell transcriptomes, given that highly correlated single cells are still effectively “collapsed” into a representative state based on the regularisation parameter?
- The authors imply that algorithm performance is affected if there is a technological mismatch between ST and reference datasets. Is Redeconve more sensitive to this than other algorithms that work with averaged expression levels and/or factors? For example, what happens in the case of mismatched chemistries – e.g. 5’ vs 3’ scRNA-Seq vs polyA vs probe-based FFPE Visium?

Minor:

- In figure 1, the authors show a very cramped violin plot showing correlations between cell type signatures obtained using different methods. This should be replaced with a more adequately spaced plot as it is impossible to read.
- “Human Brest Cancer” misspelling on line 517 of the manuscript.
- “Single-cell resolution by Redeconve enables identification of pancreatic cancer clone-specific T cell infiltration” The title of this section is a little confusing – I initially read this to mean T-cell clone, rather than cancer clone.
- Supplementary figures would benefit from clearer labelling.

Reviewer #2 (Remarks to the Author):

The work presented an approach that achieves single-cell resolution deconvolution of ST data. While the presentation and the results indicated that the mathematical definition and the quadratic programming solution were sound (albeit having some scalability issue, only applicable to thousands of cells), the work itself did not sufficiently address a critical question, why and when are achieving single-cell resolution necessary, since many cells are similar? In addition, it appeared to introduce a new interpretability issue, which ones of the thousand cell states matter and how to link them with current knowledge of cell-types/states? The authors demonstrated some ad hoc exemplar results (e.g., T cells in the PDAC Visium data) based on the “novel”/more-granular states, which were interesting. However, there were no biological validation of these results, nor did the authors provide a principal way for tusers to identify/isolate cell states of interest (non-trivial given thousands of cell states) and leverage them for farther knowledge discovery. Validation is largely at technical level. Impact of the work is unclear and likely limited due to lack of interpretability and scalability. Writing is largely acceptable, but figures need substantial improvement.

Detailed comments:

- Line 17, It will be great to delineate sequencing-based vs imaging-based technologies such as MERFISH, which are at molecule/subcellular resolution (does not require deconvolution).
- Figures are heavy but difficult to link to the points in the text. For example, line 62: “Redeconve had higher consistency with each other than mapping-based methods (Fig. 1b)”. Very difficult to

see/judge the consistency in Fig. 1b. "indicating the relative superiority and robustness of deconvolution-based methods". Over-generalized statement based on results from one dataset.

- The supplementary figures are not labeled in the individual PDFs (or filenames). Thus, I cannot match them with the text.
- Fig. 1d only has one Redeconve result. Does it correspond to "with suitable reference" (or "no suitable reference") described in the text?
- The cosine similarity results and the Sankey diagrams in various figures are largely expected if the method works as promised. Therefore, these figures only serve as technical validation and do not yield any interesting biological insights.

Reviewer #3 (Remarks to the Author):

Summary:

This work uses a relatively simple algorithm to deconvolute the low spatial resolution spatial transcriptomics data (such as 10x Visium data) by leveraging the reference scRNA-Seq or snRNA-Seq data. There are known caveats in such type of work, including the non-linear relations between scRNA-seq/snRNA-seq and spatial data, non-normal distribution of gene expressions (dropout-related zero-inflated negative binomial distribution, for example), singular matrix due to the similarities of gene expressions among similar cells, and the fundamental difference in capturing genes by in situ (10x Visium, e.g.) and cell-dissociation- based biotechnologies. With a simple linear model, this method achieves impressive performance in terms of accuracy compared with peers, which is a surprise. The unique strength of this work is the direct use of reference cells without annotations – which enables a series of downstream data analyses to identify single-cell-level spatial heterogeneity. The major concerns are 1) the significance of the work has been undermined due to the new trend of commercially available single-cell spatial profiling platforms; 2) lack of ground truth for some essential performance metrics – I provide some suggestions for ground truth; 3) the results are not complete; 4) the source code is not accessible, thus I cannot reproduce the results or check whether the codes and the algorithm are consistent. Once necessary ground truth, all results, and the source codes are provided, I am glad to re-evaluate the manuscript.

Major concerns:

1. Commercial single-cell spatial omics platforms such as NanoString CoxMx, Vizgen MERSCOPE, and the 10x Genomics Xenium are emerging. The significance of deconvoluting low spatial resolution data such as 10x Visium data is likely to decrease significantly. For example, the preprint about and the data from the NanoString CoxMx have been made publicly available since last Nov (<https://www.biorxiv.org/content/10.1101/2021.11.03.467020v1> and <https://nanosttring.com/products/cosmx-spatial-molecular-imager/ffpe-dataset/>). The Vizgen MERFISH/MERSCOPE preprints and datasets have also been available since this March (<https://www.biorxiv.org/content/10.1101/2022.03.04.483068v1> and <https://info.vizgen.com/data-release-program>). Researchers start to receive their single-cell spatial transcriptomics data from these providers. Deconvoluting cell compositions are no longer necessary for such types of data. Establishing the significance of this work under the context of the current transition from low-spatial-resolution platforms to such commercially available high-spatial-resolution platforms might be necessary.
2. Thorough performance evaluations on representative datasets for essential performance metrics are necessary. Ground truth and quantitative evaluations are missing for some results. In details:
 - a) Unless technically challenging, all performance metrics should be provided for all datasets to avoid concerns of cherry-picking the benchmarking results. Such results can be shown in supplement materials.
 - b) Fig 1b: it looks like only the breast cancer dataset is used. Could the authors provide the same analysis for other datasets?
 - c) Fig 1 e: Since one goal of this tool is to exactly predict how many cells are on each spot, and since the exact number of cells on each spot can be determined at relatively high accuracy from the H&E images, a thorough comparison from predicted cell numbers at each spot and the number of cells identified on each spot from the H&E image can be helpful. Currently, only the results from the mouse brain are shown (Fig 1e and Suppl Fig 1). Brain tissues are not the most representative

example, due to the cell dissociation challenges and the nature of the reference data (snRNA-Seq), as recognized by the authors and demonstrated in Fig 1 d. Evaluating the model performance using these essential metrics on other tissues is preferable.

d) Fig 1 f: human lymph node dataset is only used for these metrics but not in other sub-figures in Fig 1.

e) Fig 1 f: runtimes on other datasets could provide a comprehensive understanding of the model performance in different scenarios. From Fig 1d, it looks that the Human Testis dataset is suitable for testing the performance on a large number of spots. 10x Visium might be suitable for performance evaluation on a large number of genes.

f) Fig 1 f: The increase of runtime with respect to the increase in the number of cells will be informative for researchers, as the current data analysis is moving toward large datasets. Whether parallel computation was used for Redeconve (and if used, the number of threads) was not specified. Supplement Figure 9 does not provide the runtime for Redeconve.

g) Fig 3d: would it be possible to provide quantitative measures to support the claim that "Redeconve and CARD showed relatively clear boundaries of tissue regions in accordance with histological analysis" (Line 111 – 112)?

h) Fig 4a: ground truth is missing. It is relatively easy to distinguish lymphocytes in H&E images. Could the authors quantitatively examine the H&E images at the corresponding spots to confirm whether the predicted high-T-cell spots are colocalized with more lymphocytes? The original H&E image on GEO (GSM3036911_PDAC-A-ST1-HE.jpg) is of high resolution (42,309 by 36,028 pixels) and is sufficient for identifying lymphocytes by cell morphologies. Meanwhile, evidence is needed to support the claim that "reported T cells in almost all spots" "is not so reasonable in such a cancer tissue" (line 118). Actually, modest T cell infiltration in PDAC is common, and the original paper of the PDAC dataset (PMID: 31932730) suggested uniform distributions of T cells across all regions (Fig 2h in PMID: 31932730).

i) Detailed results supporting major conclusions/figures are expected as supplement tables. For example, the deconvolution results of CellTrek are very different from the rest methods (the peak distribution is at 0) in Fig 1 b, and the accuracy of Redconve on breast cancer (Fig 1 c) and lymph node (Supplement Figure 3) are dramatically better than other methods. Thus the deconvolution results (not only the metrics/scores) will allow the audience to better understand this. Fig 5 only shows results on one dataset – the similar analysis results of other datasets should be provided as supplement files too.

3. The use of perplexity/information entropy for performance evaluation of the predicted complexity at each spot (Fig 2b) is an interesting idea. As the authors mentioned, the performance should not be simply evaluated by higher or lower perplexity scores, but by biological meanings.

a) Just wonder whether biological ground truth for some of the datasets can be provided to interpret the results in Fig 2b. Again, publicly released single-cell spatial data can be used to mimic the spot-based data and thus can provide some degree of ground truth for the distribution of the perplexity score.

b) Some biological insights into different perplexity levels would be helpful. For example, biologically, what does a perplexity score of 0 – 10 mean, and what does a score of 40 – 120 or about 175 mean. Without such insights, it is hard to tell whether the current claim, "But cell2location reported extremely high perplexity for most spots, indicating high false positive rate (Fig. 2b)", is valid.

4. The scRNA-Seq or snRNA-Seq data are used for deconvoluting spatial transcriptomics data. There are fundamental differences in the capturing of gene expressions between in situ transcriptomics data (such as 10x Visium) and cell dissociation based (such as 10x scRNA-seq) or cell nuclei dissociation-based transcriptomics data (such as 10x snRNA-seq). The mRNA expression patterns in cell plasma and cell nuclei are different, and such difference is cell-type-specific. During the cell harvest, there are tissue- and cell-type-specific loss of cell plasma or cells. For example, due to the cellular anatomy of the brain tissue, it is challenging to dissociate neuron cells without significantly losing cell plasma and losing cells. Another example is that the cell nuclei transcriptomics data (scRNA-seq data E-MTAB-11114 and E-MTAB-11115) are used to deconvolute spatial transcriptomics data generated from the intact brain tissue brain. Further analysis of the impact of such differences in the model performance might be helpful. For example, are some residuals/differences between the predicted and observed expression patterns at some spots due to this? Whether the immune-cell-rich spots see better prediction results? And whether the single-cell spatial transcriptomics data be more suitable than scRNA-seq and snRNA-seq data for deconvolution.

5. Scalability in terms of runtime and accuracy. Due to the heterogeneity in the population, there are ongoing efforts of collecting million-level reference scRNA-Seq data (as the Human Cell Atlas and the NCI Human Tumor Atlas Network) to better represent the population. Scalability of the model when the reference cells reach to sub-million or million level and the number of spots reaches to sub-million level (as the human testis slide-seq data used in this work) might be necessary for many use cases. The scalability through parallel computing by splitting spots and the impact on accuracy should be provided. Ideally, the runtime vs the number of spots, vs the number of reference cells, and vs the number of parallel threads should be provided, and the accuracy vs the number of parallel threads should also be provided.
6. Simple randomized sampling is used for selecting reference cells. Since cell types are unbalanced in the scRNA/snRNA datasets, a cluster-based selection might be more suitable, or at least a sufficient selection of cells from less frequent cell types might be necessary.
7. Spatial information of the spots is not used in training the model using spatial data. A justification will be helpful.

Minor concerns:

- Fig 1c and Fig 5b: are the crossing of edges in the Sankey diagrams necessary? What the order of the cells on the right side represents?
- Fig. 4: two cancer clones were suggested. Could the authors visualize the number of cells per spot of each clone in the spatial tissue (a similar figure such as Fig 4a – but with the color intensity representing cell numbers instead of cell proportions) please? Cells belonging to the same clone are often spatially close to each other.
- Labeling the datasets and the performance metrics directly in the figures may help the audience to read.

To reviewer #1:

Implementation:

Reconstructing single cell composition of lower than single cell resolution spatial transcriptomics data is an active area of research and currently a largely unresolved problem in computational biology. The work presented by the authors is therefore very timely and any advancements in this area would be very welcomed by the research community. Existing methods for ST deconvolution use cluster-level information to infer the composition of individual ST spots, assuming that a cluster represents a homogeneous, representative cell type. Here, the authors describe a novel method that instead deconvolutes ST spots based on transcriptomes of single cells instead. While conceptually novel and potentially a very useful approach, I have a number of comments/concerns that should be addressed before I can recommend the manuscript for publication.

We thank the reviewer for the enthusiastic comments.

Implementation:

- The implementation of the described algorithm is not particularly user friendly or well documented. Publishing this as a package with adequate documentation would be essential. I would also highly recommend providing a Seurat/scanpy wrapper as this would make the method much more accessible and more highly used.

Reply: We thank the reviewer for the constructive suggestion. Originally, we uploaded our code on CodeOcean for reviewers' analysis. Now the package is also available on github with detailed documentation (<https://github.com/ZxZhou4150/Redeconve>). We also add a wrapper to enable the direct usage of Seurat object for deconvolution analysis by Redeconve. The current version of Redeconve is written by R. We will rewrite the algorithm with Python in future as the reviewer suggested. Please check the github page for the package and documentation.

- The manuscript and the documentation provided with the code implies that the output of the deconvolution returns a matrix of estimated cell counts per spot. While testing this with a 10x Visium and paired scRNA-seq dataset, the output was neither cell counts (floats, not integers) nor proportions per spot (do not add up to one). Some clarification on this – or better documentation – is required. The results otherwise seemed largely sensible, so I don't think I have grossly mis-used this.

Reply: We thank the reviewer for the suggestion to further improve the value of our algorithm. Now we have added a new module to improve the result interpretability. This module contains two functions: one converts the

“abundance” output to proportion, and the other converts the “abundance” values to cell counts if some priori knowledge is available. Visualization of “proportion” can be found in the main text (Fig. 4d and 6c, the two spatial piechart). Visualization of “cell counts” can be found in the newly added Response Figure 1, which is highly consistent with RNA counts. Details of this newly added module as the reviewer suggested can be found in the documentation of our software package deposited in github.

Response Figure 1. Estimated number of cells per spot and number of RNA counts per spot. Average number per spot is set to 10.

- Unlike other methods where deconvolution outputs a much smaller cell type matrix, the output here generates a much larger cell by spot matrix which is not readily interpretable without additional downstream analysis, and the output cannot be visualised as a whole beyond small toy or heavily down sampled datasets. Again, expanding the current code and providing this as an analysis package with additional functionality for querying, visualising and summarising the output would go a long way to enhance the usability of this. Currently, there is quite a lot of additional legwork required from the user just to understand the basic output.

Reply: We thank the reviewer for the constructive suggestions. We have provided additional functionality for visualizing the cell type proportion of each spot, the spatial distribution of selected cell/cell type, co-localization analysis of different cell states, gene expression imputation, etc. We also applied these functions in our manuscript. Please refer to the documentation for details.

- Throughout the manuscript, the authors claim that their method is fast (“direct usage of scRNA-Seq data as reference is computationally efficient”) in comparison to others– however, in the discussion they concede that it is not very scalable. Indeed, testing deconvolution of one Visium section and a very heavily down-sampled (50 cells per cluster) scRNA-Seq reference dataset – the running time was close to 8 hours. Attempting deconvolution of a single Visium slide using a medium sized (~50k cells) reference scRNA-Seq dataset

did not finish (yet). While compared to algorithms like cell2location, which is particularly slow, this is indeed an improvement, many others do take a fraction of the time. As scalability with reference dataset size is an issue, providing some formal guidance on strategies on how to best deal with this for end users with larger datasets would be useful. I noted that there is a section in the methods regarding dataset down sampling – it would be particularly useful to see how much impact, if any, this has on the results.

Reply: We thank again for the constructive comment. As the reviewer pointed out, Redeconve exceeds other deconvolution algorithms regarding the computational efficacy although other algorithms used only tens of cell clusters as reference while Redeconve used thousands of cell states as reference. However, the development of bioinformatics tools mainly lags the development of scRNA-seq technologies. Therefore, there is still no algorithm available in the community to conduct deconvolution analysis with more than thousands of cell states as reference, although Redeconve represents a beginning of this important direction. Theoretically, the running time of Redeconve is linearly proportional to the number of genes and the number of spots, but has an $O(n^3)$ relationship with the number of cells because the core procedure is a quadratic programming model. In practice, the speed of Redeconve is extremely fast when the number of cells is around 1000 regardless of the number of genes and spots. When the number of cells increased to 5000, Redeconve can still work well because of the parallel computing regarding ST spots. If the cell number is more than 5000, we suggest applying a downsampling operation to ensure the computational efficacy. We have implemented two types of downsampling operations in the software package, of which one is random downsampling and the other is downsampling guided by user-providing clusters. Overall, considering that other deconvolution algorithms handle only tens of cell clusters but Redeconve can handle thousands of cell states, the speed of Redeconve computation is still quite fast. We will reprogram Redeconve by Python in the future to further accelerate the deconvolution speed. We have also added new speed-testing results in our revised manuscript.

Benchmarking:

- In many areas, the benchmarking against other methods is very much stacked in favour of Redeconve and the comparisons are not fair.

Reply: We have added additional datasets and metrics to comprehensively compare the performance of Redeconve and other algorithms. Please refer to our newly added figures, which again suggest the advantage of deconvolution at single-cell resolution by Redeconve.

- While I understand that one of the advantages of this method lies in the ability to map reference dataset cell states to ST spots regardless of the quality and resolution of annotations in the reference dataset, nonetheless I

do not think it is fair to feed such high level annotations as “T-cells” to other algorithms and then claim that the method proposed here can resolve the spatial location between Tregs and cytotoxic T-cells and competitors cannot. The authors should repeat this benchmarking and give the other methods a fair chance by using better annotated reference datasets – indeed, as Tregs and CD8s are such functionally different cells, any reasonable researcher would not group them together for a deconvolution task.

Reply: We appreciate the reviewer for pointing out that Redeconve does not need any previous annotation of the reference while other methods heavily depend on the quality and fineness of the annotations of the reference. Indeed, Redeconve is the first deconvolution algorithm that is independent of dataset annotation.

For the PDAC dataset, we further divided T cells into three groups including CD8 T cells, T helper cells, and Tregs, and then used the fine annotation as input to run other methods (Supplementary Fig. 24). The results suggest that Cell2location cannot distinguish the three T cell subtypes; in the results of Tangram, the abundance of T cell subsets was over-estimated for many ST spots; the results of NovoSpaRc showed a spatial distribution of T cells close to the uniform distribution, which is incorrect; DestVI and CARD fail to accurately locate T cells because the results were not consistent with the distribution of T-cell-specific genes e.g., CD3D, CD3E, and CD3G (Supplementary Fig. 23).

- To follow up on a previous point, many existing spatial deconvolution algorithms do not perform well when given a scRNA-Seq reference dataset annotated with closely transcriptionally related reference cell populations, often splitting the prediction probabilities between cell clusters with highly correlated transcriptomes. An algorithm to accurately map finer reference cell sub-populations to spatial transcriptomics data is therefore sorely needed and Redeconve approach does look promising in this regard. However, a benchmark would be needed to evaluate this – it is not explored at all at what reference dataset cluster resolution do other methods break down, but it would be a very useful reference point to have and highlight niches where the proposed method would be most applicable. Instead the authors do the opposite and restrict other methods to high level annotations only.

Reply: The reviewer raised an excellent suggestion to show the advantage of single-cell deconvolution by Redeconve. During the revision, we added experiments as the reviewer suggested to illustrate this point. Given an scRNA-seq dataset, we first generated a pseudo-bulk dataset by merging the gene expression matrix of all cells into one gene expression vector. We then clustered the single cells into different numbers of groups based on the Louvain algorithm implemented in the Seurat software package with a series of resolution parameters. By using the clusters obtained from different resolution as reference for deconvolution, we evaluated the performance

changes of different algorithms. The results showed that the performance of other methods declined sharply when the cluster number increases, *i.e.*, when closely transcriptionally related reference cell populations emerge. In contrast, our algorithm Redeconve, the only single-cell deconvolution algorithm, showed superior and stable performance(Fig. 2d). We also conducted such comparison on real examples, *i.e.*, deconvolution of ST spots with different levels of scRNA-seq clusters as reference. The results again illustrated the superiority and robustness of Redeconve compared with other algorithms (Supplementary Fig. 21 and 25). In summary, the current deconvolution algorithms performed well when tens of clusters were provided as reference. But when the cluster number reaches hundreds, the performance will break down. Redeconve directly uses single cells as reference and demonstrating stable and superior performance.

- Cosine similarity is used as a performance metric. The difference between the proposed method and others appears to be quite variable dataset-to-dataset, and also between spatial regions. It is likely that this arises due to how the predicted spot transcriptomes are computed – in the case of other deconvolution algorithms, this is based on the mean expression profile of the cell population. Therefore, if the reference dataset is under-clustered, the cosine similarity will likely show the greatest divergence in those cases and is not a reflection of real inaccuracies.

Reply: We conducted the following analysis to exclude the potential impacts of clustering on the estimation of deconvolution quality. First, we clustered the reference scRNA-seq dataset with different resolution parameters (with the number of clusters ranging from 20 to 318), and conducted deconvolution analysis with all the methods we evaluated, in which the same reference was provided for all the methods. The results confirmed the superiority of Redeconve compared with other methods at every clustering resolution (Supplementary Fig.21), excluding the concern that clustering levels might impact the fairness of comparison between Redeconve and other algorithms.

The authors should also consider benchmarking their method using alternative metrics, for example using simulated single cell mixtures. Cosine similarity alone as a metric is very high level and may hide performance issues for certain cell types – a small fraction of cell type specific genes deviating in the predicted transcriptome would not have a large impact on the overall transcriptome similarity, but would have big implications for data interpretation.

Reply: Thanks for this good advice. We have added additional metrics including Pearson's correlation and RMSE to evaluate the performance of different methods on real scRNA-seq and ST datasets (Supplementary Fig.3-4). The results again confirmed the superiority of Redeconve. For deconvolution analysis of simulated pseudo-bulk data, we directly compared

the results to the ground truth to measure accuracy (Supplementary Fig. 16-18), which also support the superior performance of Redeconve. It should be noted that only cell2location has the function to estimate the absolute cell counts, and thus we included only cell2location and Redeconve in the comparison with the ground truth.

- Alternatively, the authors should consider using some of the single cell resolution spatial transcriptomics data available publicly – e.g. CosmX or Merscope – as ground truth. For example, mouse brain sections are made available by both 10x and Vizgen, and therefore it would be interesting to use histologically equivalent regions to compare deconvolution results with close to “ground truth” single cell spatial data.

Reply: Thanks for the great suggestion. Based on a 10x Genomics Xenium dataset recently published¹, we registered Human Breast Cancer Visium IF images to Xenium morphology images, then mapped those single cells in Xenium to Visium spots based on their spatial coordinates, and generated the ground truth of cell type proportions for each Visium spot. We calculated the spot-wise cosine similarity of cell type proportions between ground truth and the results generated by different methods. As shown in Fig. 3b and Supplementary Fig. 19, Redeconve achieved the highest similarities for most Visium spots.

- Why did the authors select some but not other algorithms to benchmark against? An exhaustive comparison is probably beyond the scope of this work, but some justification of the current selection would be good.

Reply: As the reviewer pointed out, an exhaustive comparison is beyond the scope of this work. These methods we chose have demonstrated superior performance in the corresponding publications and third-party evaluation papers. Therefore, we compared Redeconve with these algorithms to show how deconvolution at single-cell resolution can advance the field.

- In several instances the authors claim that their algorithm yields better delineation of spatial region boundaries. “Redeconve and CARD showed relatively clear boundaries of tissue regions in accordance with histological analysis (Fig. 3d). Meanwhile, cell2location and DestVI showed blurred boundaries, and NovoSpaRc and Tangram did not show boundaries (Fig. 3d).” This is somewhat subjective – to me, it seems that cell2location does quite well in this instance, though it is interesting to see that NovospaRc and Tangram seem to completely fail at this particular task. “Spatial pie chart showed that Redeconve produced obvious regional division, while other methods showed blurred or even no boundaries (Fig. 5c). “In this case, it also looks like cell2location and desIV give quite clear regional divisions. The authors should formalise and quantify these interpretations - for example, by computing cell type composition cluster purity within histologically defined

regions.

Reply: We have now supplemented quantified verification of our statement by using the method implemented in the paper of CARD published in *Nature Biotechnology*. We first run PCA on the cell type result, then conducted Wilcoxon rank sum test among the four regions pairwise. The performance of Redeconve is indeed the best, with the higher $-\log(p\text{-values})$ obtained for most comparisons (Supplementary Fig.22).

- The amount of space dedicated to robustness comparisons is unnecessary and somewhat misleading – comparing a deterministic algorithm with non-deterministic ones in such a way is not particularly meaningful, although the poor stability of some other non-deterministic methods is interesting. In figure 2, the values on the scales should also be standardised as at first glance, given the small axis labels, this gives a false impression of much poorer stability of cell2location than it actually is.

Reply: We thank the reviewer for the suggestions. We have deleted the robustness comparison from the revised manuscript and standardized the axis labels.

- Comparisons attempting to use other methods with single cell input is interesting, but also ultimately not particularly useful, as these algorithms were not designed to work that way .

Reply: We added this comparison to illustrate that deconvolution at single-cell resolution cannot be achieved simply by increasing the granularity of the clustering of scRNA-seq data, which actually requires additional algorithmic innovation as implemented in Redeconve.

- “Interestingly, T cell 8 and 35 were mainly co-localized with cancer cells, indicating dispersed cancer cells outside the cancer region, which is missed by other methods”. There is no formal comparison done between Redeconve and other methods on how well their predictions are able to reconstruct co-localisation networks. This statement is based on the lack of inference of T-cell subpopulations by other methods, however as mentioned previously, this is based on the artificially restricted reference dataset annotation level. A comparison of cell co-localisation networks would be particularly interesting, especially as other methods tend to falsely infer co-localisation for closely related cell types.

Reply: As we demonstrated in the replies to previous comments, other methods failed to accurately infer the spatial locations of different T cell subsets. Co-localization analysis suggests that these methods even failed to discover the co-localization between CD8 T cells and cancer cells except Tangram. Although Tangram identified the co-localization between CD8 T cells and cancer cells, it did not reveal the cancer-clone specific recognition by different CD8+ T cell states (Supplementary Fig. 25).

Other:

- When focusing on specific cell predictions and case study sections, the authors should show the spatial expression of cell type specific genes as well as the signal obtained using their method as a “sanity check”. This is lacking through-out the manuscript. For instance, the authors imply that other algorithms erroneously predict some T-cell presence in the majority of spots in the cancer tissue slide, while Redeconve yields more localised predictions. Showing whether this corresponds to spatial expression of T-cell specific genes such as CD3D, etc. would go a long way towards convincing the reader this is indeed the truth.

Reply: We have added cell type-specific genes and the corresponding spatial distribution to verify the deconvolution results as the reviewer suggested (Supplementary Fig.23). The newly added figures confirmed the accuracy of Redeconve analysis.

- “We found co-localization of IgA+ plasma cells with CD8+ cytotoxic T cells, suggesting that CD8+ cytotoxic T cells may play important roles during the formation of IgA+ plasma cells. Furthermore, the co-location of IgG+ plasma cells and macrophages may indicate the roles of macrophages during the genesis of IgG+ plasma cells (Fig. 6d).” Co-localisation does not necessarily mean there is a functional role in class switching. I think these statements should be toned down or placed in context of relevant citations supporting these interpretations.

Reply: We thank the reviewer for the advice. We have now toned down our statement and added relevant citations to support our findings.

- Similarly, a lot of the more biological examples discussed in the manuscript are poorly cited throughout and lack appropriate literature context .

Reply: We thank the reviewer for the advice. We have added more citations to illustrate the context and support our findings.

- The analysis against nuclei segmentation is quite nice – it would be great to repeat it with different resolution/spot size ST platforms to evaluate whether this result is robust across a range of real world values, or merely tuned to perform well at the most common cell densities encountered with current gen ST platforms.

Reply: We used a human breast cancer dataset published recently by 10x Genomics to evaluate the accuracy and robustness of different algorithms in estimating absolute cell abundance. For this dataset, we mapped cells in the Xenium dataset to Visium spots, and generated the ground truth of cell abundance for each Visium spot. Comparison among the ground true and the results generated by different methods suggested that Redeconve has comparable performance with Tangram in estimating cell abundance and is

robust to scRNA-seq references (Fig. 3c). Since only cell2location and Tangram can be applied to estimate the absolute abundance of cells within Visium spots, we included cell2location and Tangram only in this comparison.

- The manuscript does not convey very well how the hyperparameter tuning affects the predictions. For instance, how does this affect closely related cell type localisation. Some additional exploration and guidance to the end user would be useful.

Reply: The hyperparameter is used to balance the two loss functions in our model. When a small hyperparameter is set, the regularization term will diminish and the deconvolution term will play the major role. When a large hyperparameter is set, the regularization term will play more important roles to resolve the co-linearity issue caused by closely similar cell states. We recommend users to set a small hyperparameter value when the co-linearity issue is weak and a large hyperparameter value when the co-linearity issue is severe. To practically verify the function of hyperparameter, we applied Redeconve to the human lymph node dataset with a series of different hyperparameters. Then we calculated the deconvolution residuals (RMSE_normal) to evaluate the effect of hyperparameter (Supplementary Fig.28). The results showed that an optimal hyperparameter can enhance the deconvolution precision in addition to avoiding co-linearity caused by closely similar cell states. The hyperparameter would also affect the number of cell states selected in the result. A bigger hyperparameter would lead to more cell states selected (Supplementary Fig.29). With a hyperparameter of 1e04, more T cells were detected than a hyperparameter of zero in the PDAC dataset (Supplementary Fig. 30). Considering the distribution of CD3+ cells (Shown in supplementary Fig. 23), this example clearly illustrates how the hyperparameter enables biological discovery.

- Following on from previous point, is Redeconve conceptually really mapping single cell transcriptomes, given that highly correlated single cells are still effectively “collapsed” into a representative state based on the regularisation parameter?

Reply: The conceptual idea of our model is to find a biologically meaningful way to solve the collinearity problem, not the “collapse” issue. Actually, due to the high accuracy of regression algorithm, in most cases it can distinguish two relatively similar cells. But with the increase of cell and gene number, chances increase that some cells are extremely similar, in which case traditional regression model would encounter the collinearity problem, which will result in uncertainty of the regression coefficients. Redeconve is designed to stabilize the uncertainty by introducing a regularization term that requires similar cells have similar coefficients. Therefore, Redeconve can conduct single-cell deconvolution in most cases by the deconvolution term and result in a stable and biologically meaningful solution when extremely similar cells exist,

consistent with evaluations in our manuscript.

- The authors imply that algorithm performance is affected if there is a technological mismatch between ST and reference datasets. Is Redeconve more sensitive to this than other algorithms that work with averaged expression levels and/or factors? For example, what happens in the case of mismatched chemistries – e.g. 5' vs 3' scRNA-Seq vs polyA vs probe-based FFPE Visium ?

Reply: We have performed further evaluations to evaluate the impact of different reference (Fig. 3d and Supplementary Fig. 20). The result showed that Redeconve generated more stable results while other methods were more sensitive to reference selection.

Minor:

- In figure 1, the authors show a very cramped violin plot showing correlations between cell type signatures obtained using different methods. This should be replaced with a more adequately spaced plot as it is impossible to read.

Reply: The initial aim of this violin plot is to illustrate the overall consistency of “deconvolution-based” methods. We have repeated such evaluation on other datasets, and verified the statement. During revision, we replaced the violin plot by heatmap plots to show the median Spearman’s correlation of cell type proportions (Fig. 1a and Supplementary Fig. 1), which should be more comprehensible.

- “Human Brest Cancer” misspelling on line 517 of the manuscript.

Reply: Sorry for this typo error. We have corrected it in our revised manuscript.

- “Single-cell resolution by Redeconve enables identification of pancreatic cancer clone-specific T cell infiltration” The title of this section is a little confusing – I initially read this to mean T-cell clone, rather than cancer clone.

Reply: Thanks for the reminding. We have added a dash between “cancer” and “clone” to prevent potential misleading.

- Supplementary figures would benefit from clearer labelling.

Reply: Thanks. We have added more labels to enhance the readability of our supplementary figures.

To reviewer 2:

The work presented an approach that achieves single-cell resolution deconvolution of ST data. While the presentation and the results indicated that the mathematical definition and the quadratic programming solution were sound (albeit having some scalability issue, only applicable to thousands of cells), the work itself did not sufficiently address a critical question, why and when are achieving single-cell resolution necessary, since many cells are similar? In addition, it appeared to introduce a new interpretability issue, which ones of the thousand cell states matter and how to link them with current knowledge of cell-types/states? The authors demonstrated some ad hoc exemplar results (e.g., T cells in the PDAC Visium data) based on the “novel”/more-granular states, which were interesting. However, there were no biological validation of these results, nor did the authors provide a principal way for users to identify/isolate cell states of interest (non-trivial given thousands of cell states) and leverage them for farther knowledge discovery. Validation is largely at technical level. Impact of the work is unclear and likely limited due to lack of interpretability and scalability. Writing is largely acceptable, but figures need substantial improvement.

Reply: We thank the reviewer for the enthusiastic comments on the technological features of our algorithm, which achieves single-cell resolution deconvolution of ST data. For the necessity of deconvolution at single-cell resolution, we want to emphasize the heterogeneity of immune cells, which play critical roles in almost all types of diseases. Because of the huge heterogeneity of immune cells, a lot of immune questions are still open. Deconvolution at single-cell resolution will unlock new molecular and cellular mechanisms of immune-related phenomena as we exemplified in our manuscript. Single-cell RNA-seq has been widely applied in tumor, immune, neurological, and developmental studies, huge phenotypic heterogeneity is revealed for almost all fields. The conventionally defined cell types/states are generally over-simplified. These observations motivate us to develop deconvolution algorithms with single-cell resolution, which is also the goal of previously published deconvolution algorithms. Because the validation experiments of our newly derived insights involve establishment of new animal models, which exceeds the capability of our lab and the scope of this manuscript, we only illustrate how single-cell-resolution deconvolution unlocks the generation of novel insights from ST data.

After our single-cell deconvolution, the cell types/states information can be employed again to help interpret the results. Our innovation is unlocking deconvolution analysis from cell types/states. Following the reviewer’s suggestion, we have added a plotting function to help users to find cells of interest. Taken the PDAC dataset as an example, as shown in Response Fig. 2, the plot shows that cancer clone B #38 exhibits high spatial variance while

cancer clone A #117 was relatively more homogeneous across spots. This plot may help users to gain more insights when applying Redeconve.

Response Fig. 2. Mean vs. standard deviation of PDAC dataset. Grey line is the fitting line. 10 outliers most far away from the line were highlighted.

Detailed comments:

- Line 17, It will be great to delineate sequencing-based vs imaging-based technologies such as MERFISH, which are at molecule/subcellular resolution (does not require deconvolution).

Reply: Sequencing-based and imaging-based methods have distinct technological features. Imaging-based method is featured by higher resolution (usually < one cell) and low gene throughput (typically tens to hundreds of genes). Therefore, the issue of “deconvolution” does not apply to imaging-based methods. Because of the low gene throughput of imaging-based methods, the discovery potential of these technologies is limited. In contrast, sequencing-based methods are featured by high gene throughput (whole-transcriptome wide) but low resolution (multiple cells per spot), and thus have high discovery potential but need deconvolution analysis. We have revised our statement to further clarify this point.

- Figures are heavy but difficult to link to the points in the text. For example, line 62: “Redeconve had higher consistency with each other than mapping-based methods (Fig. 1b)”. Very difficult to see/judge the consistency in Fig. 1b. “indicating the relative superiority and robustness of deconvolution-based

methods”. Over-generalized statement based on results from one dataset.

Reply: The aim of this violin plot is to illustrate the consistency of “deconvolution-based” methods. We have evaluated this point on other datasets and verified the statement by a more readable figure (Fig. 1a and Supplementary Fig. 1).

- The supplementary figures are not labeled in the individual PDFs (or filenames). Thus, I cannot match them with the text.

Reply: We have updated all figures and the labels to further improve the readability.

- Fig. 1d only has one Redeconve result. Does it correspond to “with suitable reference” (or “no suitable reference”) described in the text?

Reply: Fig. 1d includes four ST datasets with separate references. The two ST datasets of the top subplots had suitable scRNA-seq reference, and obtained better deconvolution results. The two ST datasets of the bottom subplots had only single-nucleus RNA-seq data available for reference, and the deconvolution results were inferior to the top two subplots.

- The cosine similarity results and the Sankey diagrams in various figures are largely expected if the method works as promised. Therefore, these figures only serve as technical validation and do not yield any interesting biological insights.

Reply: These figures were used to show the technical outperformance of Redeconve, which is the foundation of novel biological insights.

To reviewer 3:

Summary:

This work uses a relatively simple algorithm to deconvolute the low spatial resolution spatial transcriptomics data (such as 10x Visium data) by leveraging the reference scRNA-Seq or snRNA-Seq data. There are known caveats in such type of work, including the non-linear relations between scRNA-seq/snRNA-seq and spatial data, non-normal distribution of gene expressions (dropout-related zero-inflated negative binomial distribution, for example), singular matrix due to the similarities of gene expressions among similar cells, and the fundamental difference in capturing genes by in situ (10x Visium, e.g.) and cell-dissociation- based biotechnologies. With a simple linear model, this method achieves impressive performance in terms of accuracy compared with peers, which is a surprise. The unique strength of this work is the direct use of reference cells without annotations – which enables a series of downstream data analyses to identify single-cell-level spatial heterogeneity. The major concerns are 1) the significance of the work has been undermined due to the new trend of commercially available single-cell spatial profiling platforms; 2) lack of ground truth for some essential performance metrics – I provide some suggestions for ground truth; 3) the results are not complete; 4) the source code is not accessible, thus I cannot reproduce the results or check whether the codes and the algorithm are consistent. Once necessary ground truth, all results, and the source codes are provided, I am glad to re-evaluate the manuscript.

Reply: We thank the reviewer for the enthusiastic comments on the innovation of our algorithm and the potential impacts. We have completely addressed all the four concerns during revision, which are briefly summarized as follows.

First, although the development of commercially available spatial profiling platforms is fast, the high cost of these technologies limits their wide application. Single-cell deconvolution analysis enables spatial investigation of large cohorts or even population-based studies. Moreover, the current single-cell spatial technologies rely on training-based cell segmentation, which is biased while single-cell deconvolution analysis is unbiased and training-free.

Second, we have added intensive evaluations based on both simulation and experimentally-derived ground truth to demonstrate the reliability of Redeconve. The results all support the superiority of Redeconve compared with other algorithms.

Third, we have applied all the metrics to all ST datasets included in our manuscript to comprehensively illustrate the superiority of Redeconve.

Fourth, we released our codes on Codeocean for repeatability of our analysis in the initial submission. We have now released our code on GitHub as well as a detailed documentation (<https://github.com/ZxZhou4150/Redeconve>). The consistency between our algorithm and codes is ensured.

Major concerns:

- b. Commercial single-cell spatial omics platforms such as NanoString CoxMx, Vizgen MERSCOPE, and the 10x Genomics Xenium are emerging. The significance of deconvoluting low spatial resolution data such as 10x Visium data is likely to decrease significantly. For example, the preprint about and the data from the NanoString CoxMx have been made publicly available since last Nov (<https://www.biorxiv.org/content/10.1101/2021.11.03.467020v1> and <https://nanosttring.com/products/cosmx-spatial-molecular-imager/ffpe-dataset/>). The Vizgen MERFISH/MERSCOPE preprints and datasets have also been available since this March (<https://www.biorxiv.org/content/10.1101/2022.03.04.483068v1> and <https://info.vizgen.com/data-release-program>). Researchers start to receive their single-cell spatial transcriptomics data from these providers. Deconvoluting cell compositions are no longer necessary for such types of data. Establishing the significance of this work under the context of the current transition from low-spatial-resolution platforms to such commercially available high-spatial-resolution platforms might be necessary.

Reply: While the commercial emergence of CosMx, Xenium and MERFISH is important in spatial analysis, these technologies are limited by low gene throughput, with only hundreds of customized genes detected. In contrast, sequencing-based technologies such as 10x Genomics Visium are whole transcriptome-wide. Therefore, the discovery potential of CosMx, Xenium, and MERFISH is unparallel to whole transcriptome-wide spatial technologies. Single-cell deconvolution analysis will further unlock the potential of such technologies. In fact, CosMx, Xenium, and MERFISH also suffer from a critical issue, *i.e.*, cell segmentation, which now depends on deep learning-based image processing. Single-cell deconvolution analysis is also applicable to ST data derived from CosMx, Xenium, and MERFISH platforms by merging subcellular spots into bins, which is independent of image processing and unbiased. In addition, single-cell deconvolution analysis can be further extended to bulk RNA-seq data, which enables population-scale studies. Overall, single-cell deconvolution analysis is of high significance to a wide range of applications.

2. Thorough performance evaluations on representative datasets for essential performance metrics are necessary. Ground truth and quantitative evaluations are missing for some results. In details:

- a) Unless technically challenging, all performance metrics should be provided for all datasets to avoid concerns of cherry-picking the benchmarking results. Such results can be shown in supplement materials.

Reply: We have applied all metrics to all datasets included in our manuscript.

The result can be found in Supplementary Fig.3-4. All results illustrate the superiority of Redeconve compared with other algorithms.

b) Fig 1b: it looks like only the breast cancer dataset is used. Could the authors provide the same analysis for other datasets?

Reply: The same analysis was conducted on all the datasets. Please see Fig. 1a and Supplementary Fig. 1.

c) Fig 1 e: Since one goal of this tool is to exactly predict how many cells are on each spot, and since the exact number of cells on each spot can be determined at relatively high accuracy from the H&E images, a thorough comparison from predicted cell numbers at each spot and the number of cells identified on each spot from the H&E image can be helpful. Currently, only the results from the mouse brain are shown (Fig 1e and Suppl Fig 1). Brain tissues are not the most representative example, due to the cell dissociation challenges and the nature of the reference data (snRNA-Seq), as recognized by the authors and demonstrated in Fig 1 d. Evaluating the model performance using these essential metrics on other tissues is preferable.

Reply: Since ST datasets with nuclei counts available are extremely rare, we only included one dataset in our original manuscript. During the revision, 10x Genomics released a ST dataset of human breast cancer, which has both Xenium and Visium data available. This dataset provides an additional dataset with ground truth for us to evaluate the performance of all methods on estimating absolute cell abundance. In this dataset, we manually aligned Xenium to Visium coordinates, and mapped cells in Xenium to spots in Visium, thus generating the “ground truth” of cell abundance. The results again showed that Redeconve is precise and robust in estimating cell abundance compared with other algorithms that have the capability to estimate the absolute cell abundance (Fig. 3c).

d) Fig 1 f: human lymph node dataset is only used for these metrics but not in other sub-figures in Fig 1.

Reply: All metrics have been applied to the human lymph node dataset, which were elaborated in the latter figures.

e) Fig 1 f: runtimes on other datasets could provide a comprehensive understanding of the model performance in different scenarios. From Fig 1d, it looks that the Human Testis dataset is suitable for testing the performance on a large number of spots. 10x Visium might be suitable for performance evaluation on a large number of genes.

Reply: We have added runtime analyses for all the datasets in our manuscript. The result is showed in Fig. 1f and Supplementary Fig. 6. The performance of Redeconve (without GPU acceleration) is the best among deconvolution-based algorithms.

f) Fig 1 f: The increase of runtime with respect to the increase in the number of cells will be informative for researchers, as the current data analysis is moving toward large datasets. Whether parallel computation was used for Redeconve (and if used, the number of threads) was not specified. Supplement Figure 9 does not provide the runtime for Redeconve.

Reply: Theoretically, the running time of our core procedure is linearly proportional to the number of genes and the number of spots, but cubically proportional to the number of cells because of the quadratic programming nature of deconvolution. Therefore, parallel computation is implemented in Redeconve for different ST spots and the speed can be accelerated by increasing the number of cores. In practice, the speed is extremely fast when the number of cells is ~1000 regardless of the number of genes and spots. When the cell number is very large (e.g., 100,000 or more cells), a subsampling procedure (random or supervised by cell types) is implemented in Redeconve to ensure the computational speed. Details about parallel computing with Redeconve can be found in “Methods” part. Supplementary Figure 9 (Supplementary Figure 14 now) illustrates the runtime difference of DestVI and cell2location when using cell type (CT) and single cell (SC) input. It does not need to contain Redeconve result. We have added a figure, Supplementary Figure 6, to show the runtime comparison of Redeconve with other methods.

g) Fig 3d: would it be possible to provide quantitative measures to support the claim that “Redeconve and CARD showed relatively clear boundaries of tissue regions in accordance with histological analysis” (Line 111 – 112)?

Reply: We have added quantitative evaluation of this statement by the way used by the CARD paper. We first run PCA on the cell type result, then done Wilcoxon rank sum tests among the four regions pairwise. The results suggest that Redeconve revealed relatively clearer boundaries similar to histological analysis (Supplementary Fig.22).

h) Fig 4a: ground truth is missing. It is relatively easy to distinguish lymphocytes in H&E images. Could the authors quantitatively examine the H&E images at the corresponding spots to confirm whether the predicted high-T-cell spots are colocalized with more lymphocytes? The original H&E image on GEO (GSM3036911_PDAC-A-ST1-HE.jpg) is of high resolution (42,309 by 36,028 pixels) and is sufficient for identifying lymphocytes by cell morphologies. Meanwhile, evidence is needed to support the claim that “reported T cells in almost all spots” “is not so reasonable in such a cancer tissue” (line 118). Actually, modest T cell infiltration in PDAC is common, and the original paper of the PDAC dataset (PMID: 31932730) suggested uniform distributions of T cells across all regions (Fig 2h in PMID: 31932730).

Reply: It is incorrect that T cells were uniformly distributed in PDAC. We have

two pieces of evidence: 1) There are only 25 T cells (CD3+) out of all 1926 cells in the paired scRNA-seq reference. Since the scRNA-seq data and spatial transcriptomics are paired, we believe that cell proportion in scRNA-seq data reveals the cell composition in spatial transcriptomics to some extent, namely there are less likely to be so many T cells. 2) We have checked the spatial expression of T-cell-specific genes (CD3D, CD3E, CD3G), and the result showed that these genes are not evenly expressed but with some localized pattern as Redeconve showed (Supplementary Fig.23), consistent with the fact that PDAC is a cold-tumor type. Fig 2h in PMID: 31932730 was an enrichment analysis based on clusters, and thus did not suggest a uniform distribution of T cells. While the H&E image is of high resolution, the analysis belongs to pathology and thus exceeds the scope of our professional background. Although AI-based methods are being developed to improve the automaticity of pathological images, it is now still a hard task for non-pathologists. Therefore, we did not include H&E analysis in the current revision.

i) Detailed results supporting major conclusions/figures are expected as supplement tables. For example, the deconvolution results of CellTrek are very different from the rest methods (the peak distribution is at 0) in Fig 1 b, and the accuracy of Redconve on breast cancer (Fig 1 c) and lymph node (Supplement Figure 3) are dramatically better than other methods. Thus the deconvolution results (not only the metrics/scores) will allow the audience to better understand this. Fig 5 only shows results on one dataset – the similar analysis results of other datasets should be provided as supplement files too .
Reply: We have provided all the deconvolution results as additional tables of our revised manuscript. The results contain predicted cell/cell-type abundances/proportions of Redeconve and alternative methods across all datasets and are available at https://renxw.cpl.ac.cn/files/Redeconve/Additional_Table.zip.

3. The use of perplexity/information entropy for performance evaluation of the predicted complexity at each spot (Fig 2b) is an interesting idea. As the authors mentioned, the performance should not be simply evaluated by higher or lower perplexity scores, but by biological meanings.

a) Just wonder whether biological ground truth for some of the datasets can be provided to interpret the results in Fig 2b. Again, publicly released single-cell spatial data can be used to mimic the spot-based data and thus can provide some degree of ground truth for the distribution of the perplexity score .

Reply: We summarized some insights about perplexity, please see below.

b) Some biological insights into different perplexity levels would be helpful. For example, biologically, what does a perplexity score of 0 – 10 mean, and

what does a score of 40 – 120 or about 175 mean. Without such insights, it is hard to tell whether the current claim, “But cell2location reported extremely high perplexity for most spots, indicating high false positive rate (Fig. 2b)”, is valid .

Reply: Mathematically, the expression of perplexity for distribution p is

$$perp = 2^{H(p)} = 2^{-\sum_{i=1}^I p(i) \log_2 p(i)}$$

Where $H(p)$ is Shannon entropy, $p(i)$ is the probability of state i . When p is a uniform distribution (namely $p(i)$ is a constant, $\frac{1}{I}$, for all i), we can know by simple calculation that the perplexity equals to the number of states I . This means that perplexity can reveal the number of states when the distribution is uniform. For other distributions, perplexity can also approximately represent the number of states, “Number of states” in the setting of single-cell deconvolution refers to “number of cell states (or types)”. Namely, the perplexity of each spot can approximately represent the number of cell states/types occurred in this spot. We have verified this statement on both simulated and real datasets that perplexity showed good consistency with number of non-zero cell types/states in the result, but poor consistency with absolute cell abundance (Supplementary Fig.31 and Supplementary Table 3). Therefore, a perplexity score of “0 – 10”, “40 – 120”, and “about 175” means there are 0 – 10, 40 – 120, and about 175 cell states in the corresponding spots, respectively. The result of cell2location is with high perplexity, namely cell2location indicated 100+ cell states in a single spot, suggesting that cell2location may erroneously assign more cell states to spots they do not belong to, namely “high false positive rate”.

4. The scRNA-Seq or snRNA-Seq data are used for deconvoluting spatial transcriptomics data. There are fundamental differences in the capturing of gene expressions between in situ transcriptomics data (such as 10x Visium) and cell dissociation based (such as 10x scRNA-seq) or cell nuclei dissociation-based transcriptomics data (such as 10x snRNA-seq). The mRNA expression patterns in cell plasma and cell nuclei are different, and such difference is cell-type-specific. During the cell harvest, there are tissue- and cell-type-specific loss of cell plasma or cells. For example, due to the cellular anatomy of the brain tissue, it is challenging to dissociate neuron cells without significantly losing cell plasma and losing cells. Another example is that the cell nuclei transcriptomics data (scRNA-seq data E-MTAB-11114 and E-MTAB-11115) are used to deconvolute spatial transcriptomics data generated from the intact brain tissue brain. Further analysis of the impact of such differences in the model performance might be helpful. For example, are some residuals/differences between the predicted and observed expression patterns at some spots due to this? Whether the immune-cell-rich spots see better prediction results? And whether the single-cell spatial transcriptomics

data be more suitable than scRNA-seq and snRNA-seq data for deconvolution.

Reply: That is a very good question. In the mouse brain dataset, we found that immune-cell-rich spots do not show better prediction results, with *Ptprc*+ spots showing low cosine similarity between ST data and the reconstructed expression profiles. In contrast, *Mobp*+ spots (oligodendrocytes-rich spots) demonstrate higher cosine similarity (Supplementary Fig. +23), confirming the reviewer's conjecture. Because single-cell spatial transcriptomics data are limited to hundreds of customized genes, it may bring more artificial biases for deconvolution compared with scRNA-seq and snRNA-seq data.

Supplementary Fig. +23. Spatial distribution of cosine similarity, *Mobp* and *Ptprc* expression.

5. Scalability in terms of runtime and accuracy. Due to the heterogeneity in the population, there are ongoing efforts of collecting million-level reference scRNA-Seq data (as the Human Cell Atlas and the NCI Human Tumor Atlas Network) to better represent the population. Scalability of the model when the reference cells reach to sub-million or million level and the number of spots reaches to sub-million level (as the human testis slide-seq data used in this work) might be necessary for many use cases. The scalability through parallel computing by splitting spots and the impact on accuracy should be provided. Ideally, the runtime vs the number of spots, vs the number of reference cells, and vs the number of parallel threads should be provided, and the accuracy vs the number of parallel threads should also be provided.

Reply: Theoretically, the running time of our core procedure is linearly proportional to the number of genes and the number of spots, but cubically proportional to the number of cells. Because we implemented parallel computing regarding ST spots, the speed can be accelerated by increasing the number of threads. In practice, the speed is extremely fast when the number of cells is ~1000 regardless of the number of genes and spots. When the number of cells reach several or tens of thousands, the speed begin to drop because of the quadratic programming nature but still faster than existing deconvolution algorithms. The deconvolution accuracy will increase with the increase of cell numbers. This can be seen from Response Fig. 3, where the

accuracy of single-cell modes (supervised and unsupervised) is better than that of cell-type mode (CT). Therefore, a trade-off should be balanced between runtime and accuracy. The number of threads has no effect on accuracy, because we deconvolute each spot independently. We will upgrade our algorithm to further supporting atlas-level single-cell deconvolution in future.

Response Fig. 3. Cosine similarity Lineplot with Redeconve supervised, unsupervised and cell types mode. The dataset is Human Lymph Node. In supervised mode, we selected 1000 cells with stratified sampling according to cell types. In unsupervised mode, we selected 1000 cells with simple randomized sampling. In cell types mode, we used cell type expression profile to perform deconvolution. The performance of supervised mode is the best, then unsupervised, and cell types is the worst.

6. Simple randomized sampling is used for selecting reference cells. Since cell types are unbalanced in the scRNA/snRNA datasets, a cluster-based selection might be more suitable, or at least a sufficient selection of cells from less frequent cell types might be necessary.

Reply: Actually, our sampling method is composed of two types. One is random sampling, without considering the availability of cell type annotation.

The other is a stratified sampling, in which cells from each annotated cell type are sampled with a predefined probability. We have also included a “protect” functionality to guarantee that at least one cell is selected for each cell type. Please refer to the documentation of function “cell.sampling” for details.

7. Spatial information of the spots is not used in training the model using spatial data. A justification will be helpful.

Reply: We have now not included spatial information into deconvolution because of: (1) parallel computing needs; and (2) additional spatial assumption is needed to add. We will evaluate the reasonability of spatial assumptions and update our algorithm accordingly in the future.

Minor concerns:

- Fig 1c and Fig 5b: are the crossing of edges in the Sankey diagrams necessary? What the order of the cells on the right side represents?

Reply: The crossing edges are used to illustrate how the single cells are finer subdivided from those cell types. The order has no specific meaning.

- Fig. 4: two cancer clones were suggested. Could the authors visualize the number of cells per spot of each clone in the spatial tissue (a similar figure such as Fig 4a – but with the color intensity representing cell numbers instead of cell proportions) please? Cells belonging to the same clone are often spatially close to each other.

Reply: We have supplemented the figures as the reviewer suggested. Please see Fig. 5c for details.

- Labeling the datasets and the performance metrics directly in the figures may help the audience to read.

Reply: Thanks for this advice and we have improved the readability of our figures by better labeling.

1. Janesick, A. et al. High resolution mapping of the breast cancer tumor microenvironment using integrated single cell, spatial and in situ analysis of FFPE tissue. *bioRxiv*, 2022.2010.2006.510405 (2022).

Reviewer #1 (Commented by Reviewer #3):

See attached file.

Reviewer #1 Attachment on the following page

Reviewer #1 (Commented by Reviewer #3)

Comment 1: The authors have significantly improved the usability of their models. Meanwhile, it looks like the online documentations. However, the online tutorials could have been more professional and user friendly. For example, the html version of the vignette is not rendered (that is, not shown as a webpage but just source codes). No plots/figures/results are provided in the vignette. Here are some examples of the current state-of-arts for the documentation, which includes descriptions/instructions, scripts, and results, with decent formatting:

<https://scanpy-tutorials.readthedocs.io/en/latest/spatial/basic-analysis.html#>

https://github.com/AltschulerWu-Lab/MUSE/tree/master/MUSE_demo

The authors are encouraged to provide such type of vignettes or tutorials – no need to reach such professional level, though. GitHub provides a good support for Jupyter notebook for both R and Python, which could be an option. Also GitHub allows rendering html files via GitHub Pages.

Meanwhile, this could be considered as a post-acceptation requirement and no longer needs further review by reviewers.

Comment 2: down-sampling for larger scRNA datasets. It is arguably acceptable, as sampling from large or even multiple scRNA reference datasets can be practically important to mitigate the unbalanced cell types. Using C/C++ for computationally intensive components, parallel computing, GPU-based mini-batching are more realistic solutions than reprogramming the tool by Python.

Comment 3: benchmarking. The reviewer asked the authors to use metrics that can evaluate the performance of cell types with smaller number of cells, arguing that cosine similarity, which provides the global measure across all cells, is not capable for effectively detecting inaccuracies for such less abundant cell types. The authors used RMSE and Pearson's correlation at the global level, which has not addressed the reviewer's concern. Could the authors also show the performance metrics for each cell type please?

Comment 4: benchmarking: the reviewer asked the for a justification of the current selection of other algorithms to compared with. The authors' response could be more specific and sufficient. Specifically, the authors claimed that "These methods we chose have demonstrated superior performance in the corresponding publications and third-party evaluation papers". One issue is that, CytoSPACE, which is introduced and cited in the Introduction section of this manuscript, demonstrated better performance than Trangram and CellTrek (Figure 2 g in <https://www.biorxiv.org/content/10.1101/2022.05.20.488356v1.full.pdf>). However, Trangram and CellTrek instead of CytoSPACE were chosen for comparison. This is not consistent with authors' claim.

There is no need to add CytoSPACE in, but a careful and thoughtful justification of the choices of algorithms will improve this manuscript.

Comment 5: sanity check of T cell specific genes. Suppl Fig 23 showed the spatial expression of CD3, CD8, and FOXP3. CD3 is often considered as a highly specific marker for all T cells. A concern is that why most CD8+ and FOXP3+ spots are not CD3+? The response was not specific on why Suppl Fig 23 provided a solid evidence that the results are sane. The claim of “newly added figures confirmed the accuracy of Redeconve analysis” was not obvious. Actually, the sanity check apparently suggested that the results were not consistent with the current biology knowledge that CD3 is the marker of T cells.

Reviewer #2 (Remarks to the Author):

The authors did not sufficiently address my concerns on interpretability (are these high resolution states real? If so, what do they mean biologically?). An updated result was provided showing that PDAC cancer clones can have different degree of spatial variance. The example does not strongly support the benefit of the approach, because cancer clones (states) are much easier to delineate than other cell types. The experiment also lacks proper controls to prove that the results cannot be obtained using other programs.

Anyway, the impact of the work remains unclear to me and potentially limited at technical level.

Reviewer #3 (Remarks to the Author):

The authors decently addressed most of my concerns and many thanks to the authors for the great responses. There are some minor issues and hopefully the authors can further enlighten me on these issues.

Major concerns 1: could the authors summarize the reply in Introduction please? It will provide the necessary context for the audience.

Major concerns 2 (c): detecting cell nuclei from H&E images is a basic bioinformatics task and there are many automatic tools. Since this provides ground truth for the model, the reviewer recommends (but not requests – that is, optional) generating such ground truth, using it to statistically evaluate performance, and discussing the results. Meanwhile, Figure 3(c) does not provide quantitative / numeric values of the correlations and the performance difference across different models (the left first column) is hard to tell. It looks like Trigram instead of Redeconve shows the best performance. Could the authors provide numeric values of correlations for Figure 3 (c) please? The review tried to download the file https://renxw.cpl.ac.cn/files/Redeconve/Additional_Table.zip but it showed that it needed 14 hours and it was never successfully downloaded. Also could the authors provide scatter plots for the comparisons between each model and the ground truth, with each plot for the comparison of one model against the ground truth, each point in the scatter plot representing a spot, x axis representing the ground truth, and y axis representing the model result please? It is common to show individual data points to support summarized statistical metrics such as correlation scores.

Major concerns 2 (g): it is hard to tell whether Redeconve outperforms cell2location. CARD does not show good performance compared with cell2location. The means of the $-\log(p)$ of Redeconve, cell2location, and CARD are 21.22, 24.55, and 18.512, respectively. Therefore, the results do not support the claim that Redeconve and CARD show relatively clearer boundaries. I would recommend the authors revise the text according to the quantitative results in Suppl Fig 22, maybe replacing CARD with cell2location.

REVIEWER COMMENTS

Reviewer #1 (Commented by Reviewer #3)

Comment 1: The authors have significantly improved the usability of their models. Meanwhile, it looks like the online documentations. However, the online tutorials could have been more professional and user friendly. For example, the html version of the vignette is not rendered (that is, not shown as a webpage but just source codes). No plots/figures/results are provided in the vignette. Here are some examples of the current state-of-arts for the documentation, which includes descriptions/instructions, scripts, and results, with decent formatting:

<https://scanpy-tutorials.readthedocs.io/en/latest/spatial/basic-analysis.html>

https://github.com/AltschulerWu-Lab/MUSE/tree/master/MUSE_demo

The authors are encouraged to provide such type of vignettes or tutorials – no need to reach such professional level, though. GitHub provides a good support for Jupyter notebook for both R and Python, which could be an option. Also GitHub allows rendering html files via GitHub Pages.

Meanwhile, this could be considered as a post-acceptation requirement and no longer needs further

Reply: Thanks for the suggestions. We have further improved our online documentation, and now it contains instructions, scripts and results, which may further help users to use our tool to do analysis. The jupyter notebook and rendered html and pdf files are available on the GitHub page (<https://github.com/ZxZhou4150/Redeconve>). We will keep improving this package to make it more user-friendly.

Comment 2: down-sampling for larger scRNA datasets. It is arguably acceptable, as sampling from large or even multiple scRNA reference datasets can be practically important to mitigate the unbalanced cell types. Using C/C++ for computationally intensive components, parallel computing, GPU-based mini-batching are more realistic solutions than reprogramming the tool by Python.

Reply: As the sizes of scRNA-seq datasets increase exponentially, down-sampling is an effective operation to reduce data redundancy, improve computational efficacy, and control the unbalanced cell types, as the reviewer points out. To improve the computational efficiency of Redeconve, we have taken the following operations: (1) use Fortran to solve the quadratic

programming problem; (2) apply parallel computing to accelerate spatial transcriptomics data analysis at the spot level. Fortran is a programming language with efficacy comparable with or even higher than C/C++. As Python has better support for GPU computing than R, we will rewrite Redeconve in Python to further improve the efficiency with GPU acceleration. We will also try to implement the idea with the deep learning frameworks instead of quadratic programming to further enhance the performance of Redeconve in the future.

Comment 3: benchmarking. The reviewer asked the authors to use metrics that can evaluate the performance of cell types with smaller number of cells, arguing that cosine similarity, which provides the global measure across all cells, is not capable for effectively detecting inaccuracies for such less abundant cell types. The authors used RMSE and Pearson's correlation at the global level, which has not addressed the reviewer's concern. Could the authors also show the performance metrics for each cell type please?

Reply: Thanks for your explanation. During this round of revision, we added benchmarking at the cell-type level based on the PDAC dataset (Supplementary Fig. 23). For each marker gene of each cell type, we calculated Pearson's correlation to measure the spatial consistency between ST observation and reconstructed expression profiles generated by different algorithms. Then we calculated the average correlation scores across different marker genes of each cell type and plotted a heatmap for comparison. The results showed that, for most cell types (either abundant or rare in the scRNA-seq dataset), Redeconve achieved superior performance (13/20 in top one, and 20/20 in top three) (Supplementary Fig. 23a and b). We further assessed the impacts of cell type abundance. The results showed that the performances of Redeconve, cell2location, and Tangram are robust to abundance variations among cell types, while the performances of DestVI, CARD, and NovoSpaRc are positively correlated with cell type abundances ($P < 0.05$) (Supplementary Fig. 23c).

Comment 4: benchmarking: the reviewer asked the for a justification of the current selection of other algorithms to compared with. The authors' response could be more specific and sufficient. Specifically, the authors claimed that "These methods we chose have demonstrated superior performance in the corresponding publications and third-party evaluation papers". One issue is that, CytoSPACE, which is introduced and cited in the Introduction section of this manuscript, demonstrated better performance than Tangram and CellTrek (Figure 2 g in <https://www.biorxiv.org/content/10.1101/2022.05.20.488356v1.full.pdf>). However, Tangram and CellTrek instead of CytoSPACE were chosen for comparison. This is not consistent with authors' claim.

There is no need to add CytoSPACE in, but a careful and thoughtful justification of the choices of algorithms will improve this manuscript.

Reply: CytoSPACE was not included for comparison because it uses cell2location to estimate abundances/fractions of different cell types and then samples single cells according to the estimated abundance for mapping. That is, the performance of CytoSPACE in abundance estimation depends on cell2location, and thus comparison with cell2location can also show the performance of CytoSPACE. We have revised our statement for higher accuracy regarding our criteria.

Comment 5: sanity check of T cell specific genes. Suppl Fig 23 showed the spatial expression of CD3, CD8, and FOXP3. CD3 is often considered as a highly specific marker for all T cells. A concern is that why most CD8+ and FOXP3+ spots are not CD3+? The response was not specific on why Suppl Fig 23 provided a solid evidence that the results are sane. The claim of “newly added figures confirmed the accuracy of Redeconve analysis” was not obvious. Actually, the sanity check apparently suggested that the results were not consistent with the current biology knowledge that CD3 is the marker of T cells.

Reply: Compared to single-cell RNA sequencing, spatial transcriptomics technologies have more issues in contamination and drop-outs because of the extremely small volume of amplification reaction at each spot. Therefore, different mRNAs, e.g., CD3, CD8 and FOXP3 are randomly sampled in the final ST readouts, and not all T cells are finally CD3-positive. In fact, the dropout issue is also present in single-cell RNA sequencing, but more severe in spatial transcriptomics technologies.

Deconvolution analysis based on single-cell RNA sequencing data has the advantage to impute gene expression values that were dropped out by ST, because deconvolution is conducted based on thousands of genes instead of hands of marker genes. We performed such imputation by Redeconve as described in Methods, and revealed high consistency between the estimated cell abundance and the expression levels of *CD3*, *CD8* and *FOXP3* (Supplementary Fig. 26). On *CD3*-negative ST spots that had T cells indicated by Redeconve, additional molecular evidence is present and supports the existence of T cells, e.g., the expression of *IL32* and *TMSB4X*, which are specifically expressed by different T cells (Supplementary Fig. 27).

Reviewer #2 (Remarks to the Author):

The authors did not sufficiently address my concerns on interpretability (are these high resolution states real? If so, what do they mean biologically?). An updated result was provided showing that PDAC cancer clones can have different degree of spatial variance. The example does not strongly support the benefit of the approach, because cancer clones (states) are much easier to delineate than other cell types. The experiment also lacks proper controls to prove that the results cannot be obtained using other programs.

Anyway, the impact of the work remains unclear to me and potentially limited at technical level.

Reply: In this study, we used two examples to show the biological interpretability. One is the PDAC dataset. Among all the algorithms we evaluated, only our algorithm Redeconve revealed the state differences of CD8+ T cells, which are spatially caused by different cancer clones. Because cancer clone B cells have higher levels of HLA genes and interferon-stimulating genes (ISGs), the co-localized CD8+ T cells have higher levels of TCR-related responses, consistent with the common senses of T cell immunology. The other example is the human lymph node dataset. Again, among all the algorithms we evaluated, only our algorithm Redeconve revealed the co-localization of IgA+ plasma cells with cytotoxic CD8+ T cells, consistent with the in vitro results that cytotoxic CD8+ T cells can boost the production of IgA. Therefore, we believe these two examples successfully demonstrate the biological interpretability and reality of the results generated by our algorithm beyond the available tools (controls showing that our findings cannot be generated by these programs). With the wide application of spatial transcriptomics technologies, our algorithm has the potential to reveal more novel biological insights because of the high resolution.

Reviewer #3 (Remarks to the Author):

The authors decently addressed most of my concerns and many thanks to the authors for the great responses. There are some minor issues and hopefully the authors can further enlighten me on these issues.

Major concerns 1: could the authors summarize the reply in Introduction please? It will provide the necessary context for the audience.

Reply: Thanks for the suggestion. We have integrated our replies into the introduction of our main text.

Major concerns 2 (c): detecting cell nuclei from H&E images is a basic bioinformatics task and there are many automatic tools. Since this provides ground truth for the model, the reviewer recommends (but not requests – that is, optional) generating such ground truth, using it to statistically evaluate performance, and discussing the results.

Reply: Thanks for the suggestion. We have added results of algorithm evaluation based on H&E segmentation (Supplemental Fig. 5). The results showed that Redeconve has superior or comparable performances with cell2location and Tangram in estimating absolute cell abundance for the mouse brain, PDAC, and human breast cancer datasets.

Meanwhile, Figure 3(c) does not provide quantitative / numeric values of the correlations and the performance difference across different models (the left first column) is hard to tell. It looks like Tangram instead of Redeconve shows the best performance. Could the authors provide numeric values of correlations for Figure 3 (c) please?

Reply: We have added the numeric values of the correlation on Figure 3(c) now. The results showed that Redeconve and Tangram outperformed cell2location. With single-cell RNA-seq data based on FFPE, Redeconve performed as well as Tangram. With 5' and 3' single-cell RNA-seq data, Tangram performed slightly better than Redeconve because Tangram explicitly minimizes the divergence between ground truth (or total UMIs) and predicted cell abundance in its objective function. Considering that Redeconve does not use the information of ground truth (or total UMIs), such performance further confirms the superiority of Redeconve. On the mouse brain and PDAC datasets, Redeconve outperformed Tangram (Supplemental Fig. 5).

The review tried to download the file https://renxw.cpl.ac.cn/files/Redeconve/Additional_Table.zip but it showed that it needed 14 hours and it was never successfully downloaded.

Reply: The issue may be caused by our limited international Internet bandwidth. We have also uploaded these files on Google Drive now, which can be access through the following URL link: (<https://drive.google.com/file/d/1BfgljX2CYaIEISdJy4oZ5YLOzpvf92Co/view?usp=sharing>).

Also could the authors provide scatter plots for the comparisons between each model and the ground truth, with each plot for the comparison of one model against the ground truth, each point in the scatter plot representing a spot, x axis representing the ground truth, and y axis representing the model result please? It is common to show individual data points to support summarized statistical metrics such as correlation scores.

Reply: Thanks for the suggestion. We have added the scatter plots in Supplementary Fig. 20.

Major concerns 2 (g): it is hard to tell whether Redeconve outperforms cell2location. CARD does not show good performance compared with cell2location. The means of the $-\log(p)$ of Redeconve, cell2location, and CARD are 21.22, 24.55, and 18.512, respectively. Therefore, the results do not support the claim that Redeconve and CARD show relatively clearer boundaries. I would recommend the authors revise the text according to the quantitative results in Suppl Fig 22, maybe replacing CARD with cell2location.

Reply: Because the question is inherently a hypothesis-testing question, the p values should be interpreted as the likelihood whether clear boundaries exist, rather than how clear the boundaries are. We have added more labels and annotations on the figure (Supplemental Fig. 24), and indicated the hypothesis-testing results. Redeconve, CARD, and DestVI can successfully discriminate all the four pathologically-defined regions, but cell2location, Tangram and NovoSpaRc failed in several conditions. We have also revised the text accordingly. Thanks again for the advice.

1. Vahid, M.R. et al. High-resolution alignment of single-cell and spatial transcriptomes with CytoSPACE. *Nat Biotechnol* (2023).
2. Ni, Z. et al. SpotClean adjusts for spot swapping in spatial transcriptomics data. *Nat Commun* **13**, 2971 (2022).
3. Biancalani, T. et al. Deep learning and alignment of spatially resolved single-cell transcriptomes with Tangram. *Nat Methods* **18**, 1352-1362 (2021).
4. Stringer, C., Wang, T., Michaelos, M. & Pachitariu, M. Cellpose: a generalist algorithm for cellular segmentation. *Nat Methods* **18**, 100-106 (2021).
5. Palla, G. et al. Squidpy: a scalable framework for spatial omics analysis. *Nat Methods* **19**, 171-178 (2022).

Reviewer #3 (Remarks to the Author):

The authors provided acceptable replies for addressing the concerns of Reviewer 1.

The sanity check (comment 5) is not ideal, but it makes sense and is acceptable. Among the alternative markers, IL32 is fine -- it is also expressed by NK cells, though; TMSB4X is more generally expressed and thus a less specific marker.

The authors provided acceptable replies for addressing the concerns of Reviewer 1.

Reply: Thanks.

The sanity check (comment 5) is not ideal, but it makes sense and is acceptable. Among the alternative markers, IL32 is fine -- it is also expressed by NK cells, though; TMSB4X is more generally expressed and thus a less specific marker.

Reply: We agree with the reviewer that the specificity of IL32 and TMSB4X expression in T cells is not as good as the classical markers such as CD3. We thank the reviewer for thinking that “it makes sense and is acceptable”. Because of the high dropout rate of spatial transcriptomics sequencing, combinations of different genes will be more indicative than single markers to interpret cell identity within the spots. Redeconve conducts deconvolution analysis based on thousands of genes, therefore should produce more accurate estimation than single markers. In fact, NK cells are also present in the reference scRNA-seq dataset but not selected by Redeconve because T cells can better explain the measurements in these spots, further supporting the validity of our results.